# Realization of large-area ultraflat chiral blue phosphorene

Ye-Heng Song [1,2,7], M. U. Muzaffar [3,7], Qi Wang[1], Yunhui Wang[1], Yu Jia[1,4,5], Ping Cui [3], Weifeng Zhang [1,2] ✉, Xue-Sen Wang [6] ✉ & Zhenyu Zhang [3] ✉

Blue phosphorene (BlueP), a theoretically proposed phosphorous allotrope with buckled honeycomb lattice, has attracted considerable interest due to its intriguing properties. Introducing chirality into BlueP can further enrich its physical and chemical properties, expanding its potential for applications. However, the synthesis of chiral BlueP remains elusive. Here, we demonstrate the growth of large-area BlueP films on Cu(111), with lateral size limited by the wafer dimensions. Importantly, we discovered that the BlueP is characterized by an ultraflat honeycomb lattice, rather than the prevailing buckled structure, and develops highly ordered spatial chirality plausibly resulting from the rotational stacking with the substrate and interface strain release, as further confirmed by the geometric phase analysis. Moreover, spectroscopic measurements reveal its intrinsic metallic nature and different characteristic quantum oscillations in the image-potential states, which can be exploited for a range of potential applications including polarization optics, spintronics, and chiral catalysis.

The isolation of graphene[1] with remarkable electrical and mechanical properties has opened the prelude to studying other monoelemental two-dimensional (2D) materials[2-6]. As a Group-V monoelemental 2D material, black phosphorene with tunable band gap and high carrier mobility[7,8] has become one of the high-profile 2D materials recently. Apart from black phosphorene, BlueP (single-layer Blue Phosphorus, a theoretically proposed phosphorous allotrope)[9] with the same buckled honeycomb lattice as stanene[10] has also attracted increasing attention in recent years due to its wide band gap and excellent electrical properties[9]. In addition, density functional theory (DFT) computational studies[11] showed that, under tensile stress, buckled BlueP could transform into an ultraflat honeycomb lattice just like graphene with a Dirac-cone dispersion of electronic states near the Fermi energy, providing a new Dirac fermion system. Chiral materials that break the mirror symmetry and show different responses to externally incident

left-handed or right-handed polarized light have promising applications in optoelectronics[12], spintronics[13], valleytronics[14], and chiral catalysis[15]. Recently, chiral graphene has been realized in the twisted bilayer graphene system[16], which exhibits strong circular dichroism. Evidently, introducing chirality into BlueP can further enrich its physical and chemical properties and expand the application potentials.

Due to the lack of corresponding bulk materials, BlueP can only be synthesized through bottom-up methods. A recent theoretical work[17] showed that BlueP could be synthesized on GaN(001) through a half-layer-by-half-layer growth mechanism. Meanwhile, several groups reported experimental investigations on the molecular beam epitaxy (MBE) growth of nanometre-size BlueP layers on Au(111)[18], Ag(111)[19], and copper oxide substrates[20]. Recently, such growth was also attempted on Cu(111), and a buckled BlueP with Dirac cone was observed[21]. However, large-area ultraflat chiral BlueP on the Cu surface

[1]Center for Topological Functional Materials, and Henan Key Laboratory of Photovoltaic Materials, Henan University, Kaifeng 475004, China. [2]Institute of Quantum Materials and Physics, Henan Academy of Sciences, Zhengzhou 450046, China. [3]International Center for Quantum Design of Functional Materials (ICQD), and Hefei National Laboratory, University of Science and Technology of China, Hefei, Anhui 230026, China. [4]School of Materials Science and Engineering, Henan University, Kaifeng 475004, China. [5]International Laboratory for Quantum Functional Materials of Henan, Zhengzhou University, Zhengzhou 450003, China. [6]Department of Physics, National University of Singapore, 2 Science Drive 3, Singapore 117542, Singapore. [7]These authors contributed equally: Ye-Heng Song, M. U. Muzaffar. ✉e-mail: wfzhang@henu.edu.cn; phywxs@nus.edu.sg; zhangzy@ustc.edu.cn

remains elusive, which might be used to develop new types of chiral catalysts. The well-known success of large-area graphene growth on Cu foils[22,23] inspires us to tackle the challenging task.

In this study, we report the epitaxial growth of large-area ultraflat chiral BlueP on Cu(111) surface by MBE, as large as the substrate, which is limited by the wafer size. Scanning tunneling microscope (STM) results reveal that, due to the strong stretching effect for matching with the substrate, the BlueP is configured on the Cu(111) as an ultraflat honeycomb lattice rather than the buckled structure. Importantly, the ultraflat BlueP sheet develops highly ordered chiral superstructure. We propose a plausible mechanism to explain the formation of the chiral superstructure, which can be attributed to the contrary rotational stacking with substrate and the interface strain release generated by the mismatch of the heteroepitaxial lattice. The strain distribution field obtained through geometric phase analysis (GPA) on high resolution STM images further supports this inference. Scanning tunneling spectroscopy (STS) data demonstrates that the ultraflat chiral BlueP is

metallic rather than semiconducting in the case of buckled BlueP. In addition, field-emission resonance (FER) spectroscopy reveals the different characteristic quantum oscillations in the image-potential states (IPSs) above the Fermi level for adjacent chiral units, suggesting they possess different local work functions (LWFs); and they are both larger than that of Cu(111), manifesting electron transfer from the substrate. Moreover, we have achieved a reversible transformation between the chiral and achiral phases of ultraflat BlueP, which reveals a distinctive growth mode for phosphorene structures.

## Results

### Ultraflat chiral BlueP on Cu(111)

As illustrated in Fig. 1a, the epitaxial growth of BlueP was carried out by depositing phosphorus on a Cu(111) surface held at ~200 °C. The sample growth details are described in the Methods. Figure 1b displays the reflected high-energy electron diffraction (RHEED) patterns taken before (upper image) and after (bottom image) deposition of

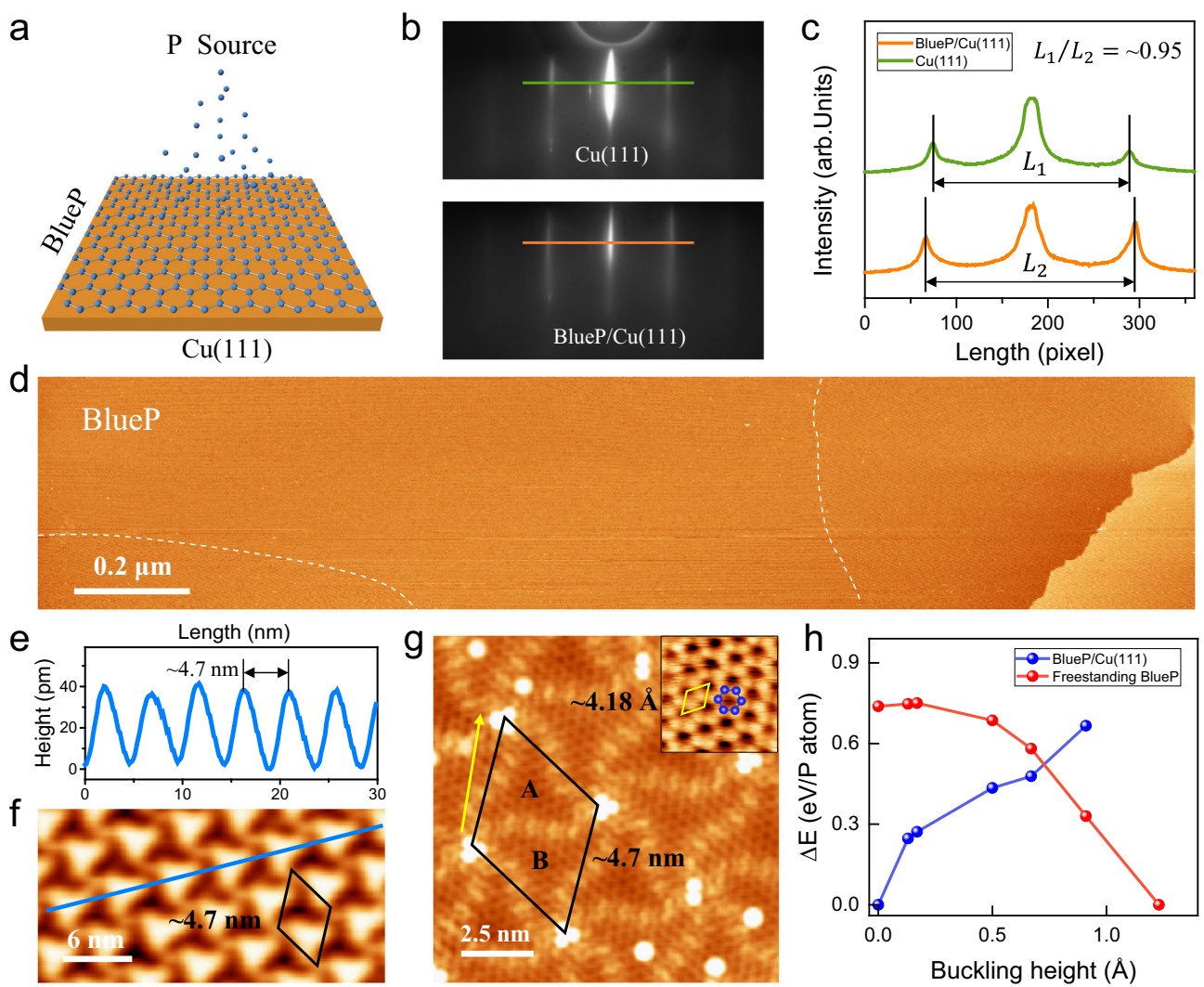

**Fig. 1 | Epitaxial ultraflat Blue phosphorene (BlueP) on Cu(111). a** Schematic diagram of ultraflat BlueP on Cu(111). **b** Reflected high-energy electron diffraction (RHEED) patterns of Cu(111) and ultraflat BlueP on Cu(111). **c** Line scan profiles along the green and orange lines as marked in (**b**), which show the distance of first diffraction streaks ($L_1$ and $L_2$) becomes larger after growth of ultraflat BlueP. **d** Large-area scanning tunneling microscopy (STM) topography image ($2.0 \times 0.4 \, \mu m^2$, $V_S = 5.0$ V, $I_t = 200$ pA, 78 K. $V_S$, bias voltage, $I_t$, tunneling current) of ~1 monolayer (ML) ultraflat BlueP, in which the dashed white lines separate two domains. **e, f** Line scan profile as marked in (**f**) and enlarged STM image

($30 \times 15 \, nm^2$, $V_S = 5.0$ V, $I_t = 200$ pA) of (**d**). **g** High-resolution STM image ($12 \times 12 \, nm^2$, $V_S = 2$ mV, $I_t = 2$ nA) of ultraflat BlueP, in which an ultraflat honeycomb lattice is clearly visible in the spiral triangle units A and B. The black diamonds in (**f**) and (**g**) mark the unit cell of superstructure. The inset is the zoom-in high-resolution STM image ($2 \times 2 \, nm^2$), in which the blue spheres mark the phosphorous atoms. The yellow diamond marks the unit cell of ultraflat BlueP. **h** Energy difference as a function of buckling height for both freestanding and supported BlueP, referenced to the most stable structure. A more positive number means less stable.

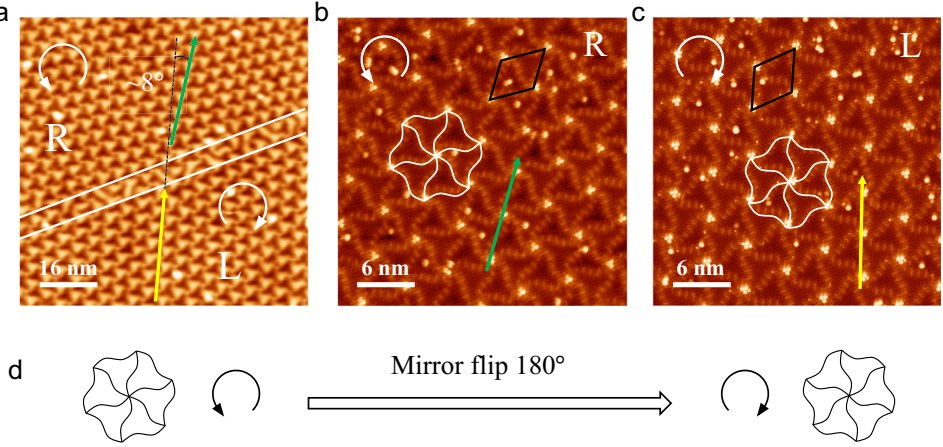

**Fig. 2 | Ultraflat BlueP with two chiral superstructures. a** STM image (80 × 80 nm², $V_S$ = 5.2 V, $I_t$ = 200 pA) that coexists two chiral domains (separated by the white lines), as marked as R and L. The green and yellow arrows guide the base vector direction of the chiral superstructures, showing an angle of ~8°. **b**, **c** Enlarged high-resolution STM images (30 × 30 nm², $V_S$ = 2 mV, $I_t$ = 2 nA) of R (right-handed chirality) and L (left-handed chirality) areas. The green and yellow arrows guide the base vector direction of R and L phases and the white fans indicate the chiral superstructures. The black diamonds mark the unite cell of the chiral superstructures. **d** Schematic diagram of chiral transformation, indicating one can obtain another chirality by performing a mirror flip operation.

~1 monolayer (ML) phosphorus on the substrate. Before deposition, the diffraction streaks of Cu(111)-$\sqrt{3} \times \sqrt{3}$ are clearly visible with incident electrons along the $[1\bar{1}0]$ direction. After deposition, the line profiles (see Fig. 1c) show that the diffraction streak spacings expanded, indicating that a new surface periodicity emerged. The measured diffraction streak spacing is 216 pixels for bare Cu, and 228 pixels for the BlueP covered sample, suggesting a ratio of lattice periods of ~ 0.95 (BlueP/Cu). Figure 1d is the large-area STM image after phosphorus deposition, where the phosphorene sheets completely cover all the terraces, and there exist two different domain areas (see Supplementary Fig. 20). Its corresponding enlarged STM image (see Fig. 1f) shows a highly ordered hexagonal superlattice with a period of ~4.7 nm, as measured from the line scan profile in Fig. 1e. The high-resolution STM image displayed in Fig. 1g reveals atomic-scale details of the two spiral triangles (denoted as unit A and unit B) in the hexagonal supercell. In the centers of units A and B, the honeycomb lattice is clearly visible, and the yellow diamond marks a unit cell. The measured lattice period is ~4.18 Å, consistent well with the RHEED results (notice the period of Cu(111)-$\sqrt{3} \times \sqrt{3}$ is 4.41 Å, and 4.18/4.41 ≈ 0.95). The lattice period obtained here is much larger than that of freestanding buckled BlueP (3.32 Å)[9], suggesting that the grown BlueP is tensilely strained to a large extent, which leads to the formation of an ultraflat honeycomb lattice, analog to graphene. Our DFT calculation results show that freestanding ultraflat BlueP is metastable (see Fig. 1h). Once introducing the substrate, we unambiguously reveal that all the P atoms of ultraflat BlueP can couple strongly with the Cu(111), which significantly reduces the relative energy compared to a buckled counterpart. The existence of phosphorus on the sample was also confirmed with semiquantitative X-ray photoelectron spectroscopy data, as displayed in Supplementary Fig. 2. Based on these results, we can conclude that we have obtained an ultraflat BlueP on Cu(111), rather than the prevailing buckled structure[21]. In particular, the large-area ultraflat BlueP can be as large as the substrate, as all the terraces are completely covered by the ultraflat BlueP sheets; the size of ultraflat BlueP is only limited by the wafer size[24].

It is noticed that the spiral triangle units show asymmetry (see Fig. 1g), and the ordered structure composed of such asymmetric units usually presents chirality[25], which lacks mirror symmetry. In Fig. 2a, two different domains (marked as R and L) are separated by a domain wall (as marked by the white lines) and linked by mirror symmetry, suggesting that they possess opposite chirality. Figure 2b, c are the high-resolution STM images obtained in the R and L regions,

respectively. For these two chiral domains, six spiral triangle boundaries interlace at every node (appear as bright trimers in Fig. 2b, c) and form an asymmetric hexagonal petal (as marked by white hexagonal petals in the images). The atomic resolution STM image (see Supplementary Fig. 13) and line scan profile (see Supplementary Fig. 4) show that these bright protrusions at the nodes are composed of three P atoms (a P trimer) located on the phosphorene layer. In R and L domains, the asymmetric hexagonal petals are like fans rotating counterclockwise (denote as right-handed chirality) and clockwise (left-handed chirality), respectively. As illustrated by the schematic diagram in Fig. 2d, one can obtain another chirality by performing a mirror flip operation. In reciprocal space, we also can define the domains' chirality referred to in previous works[26] based on the intensity of Bragg peaks (see Supplementary Fig. 6).

## The origin of chiral superstructure

It is natural to ask why the ultraflat BlueP studied here exists a chiral superstructure that is quite different from that previously predicted[9]. Our STM results reveal that the ultraflat BlueP has a lattice period of ~ 4.18 Å. Although it is stretched a lot compared to freestanding buckled BlueP (3.32 Å), it is still slightly smaller than the Cu(111)-$\sqrt{3} \times \sqrt{3}$ lattice period (4.41 Å). The mismatch of the heteroepitaxial lattice can induce a conventional moiré pattern, which has been observed in previous works of graphene on Cu(111)[27]. However, for the ultraflat BlueP studied here, we did not observe a conventional moiré pattern but a web-like spiral pattern (chiral superstructure). The absence of the conventional moiré pattern may suggest the topmost Cu(111) surface might be epitaxially matched with the ultraflat BlueP lattice, which indicates a marked reconstruction of the underlying Cu(111). The large binding energy (0.56 eV/P atom, see Supplementary Fig. 32) between P atoms and top layer Cu atoms is revealed by our calculations, which could provide a strong driving force for the reconstruction of the underlying Cu(111). Similar web-like spiral patterns have been reported in graphene/Ru(0001)[28], and graphene/Cu(111)[29]. All these systems have a common feature: the overlayer lattice periods are slightly smaller than the underlying substrate. The ultraflat BlueP on Cu(111) studied here also exhibits this feature. To minimize the surface energy, the systems would undergo a structure relaxation, resulting in a web-like spiral pattern.

Here, we propose a plausible mechanism, as shown in Fig. 3a, b, to illustrate the chiral superstructural formation process. Figure 3a is a simple model of placing a layer of compressed Cu (brown dots,

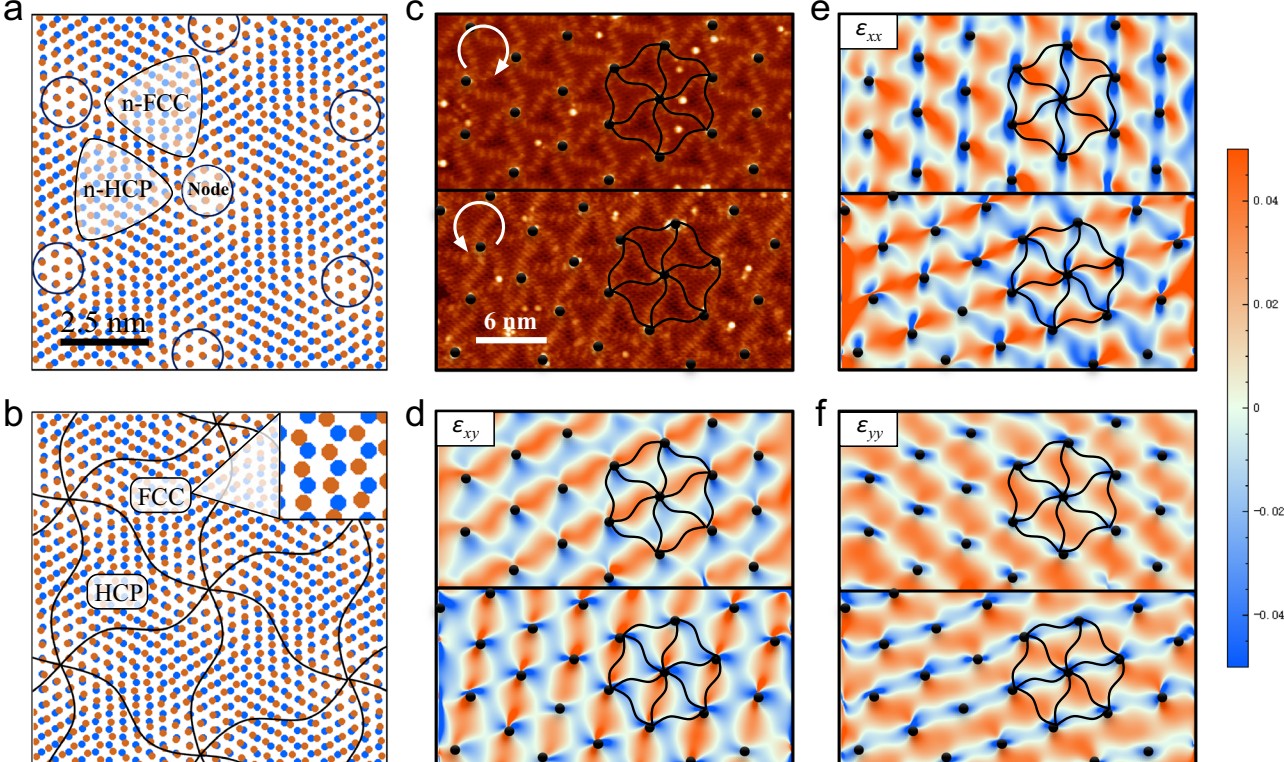

**Fig. 3 | Distribution of the strain field on the ultraflat chiral BlueP surface. a** The simulation of unrelaxed initial structure of the interface displaying the typical moiré pattern. The brown and blue dots represent Cu atoms on the surface and subsurface layers, respectively. Here, n-FCC and n-HCP represent near-Face Centered Cubic and near-Hexagonal Close Packed, respectively. **b** The proposed relaxed structure displaying the spiral bridge sites that separate local FCC/HCP-stacked areas of copper (Here, FCC and HCP represent Face Centered Cubic and Hexagonal Close Packed, respectively.). The black curves guide the domain walls. The inset shows the zoom-in image of FCC area. **c** High-resolution STM image of left-handed (upper panel) and right-handed (bottom panel) chiral regions. **d–f** The $\varepsilon_{xy}$, $\varepsilon_{xx}$ and $\varepsilon_{yy}$ strain maps of the STM images in (**c**) generated by the geometric phase analysis method (here, $\varepsilon_{xx}$ and $\varepsilon_{yy}$ are normal strains, $\varepsilon_{xy}$ is shear strains.). The black dots in (**c**)–(**f**) mark the position of nodes. The twisted hexagon marks the chiral superstructure.

matching with the ultraflat BlueP) on a Cu slab (blue dots, not compressed) with a small rotational angle, showing a conventional moiré pattern induced by lattice mismatch. Based on the relative positions of the top and bottom layer atoms, the surface can be classified into four regions: node regions, near-Face Centered Cubic (n-FCC) regions, near-Hexagonal Close Packed (n-HCP) regions, and the border regions where surface Cu atoms sit at near-bridge sites, as marked in Fig. 3a. To minimize the surface energy, the structure relaxes via atomic displacement. Under appropriate lattice mismatch and twisted angle, the n-FCC and n-HCP regions transform into FCC and HCP regions (called chiral units A and B, respectively), separated by a web-like network of borderlines (see Fig. 3b). The similar chiral superstructure has been reported in previous works[29,30]. Depending on the sign of rotational angle, the relaxed structure forms different chirality (see Supplementary Figs. 7 and 8). To verify our model, we carefully analyzed the strain distribution field of the high resolution STM images (Fig. 3c) with the help of GPA, which has been used to analyze the lattice distortion in transmission electron microscopy (TEM) and STM observations[31,32]. The tensor components $\varepsilon_{xy}$, $\varepsilon_{xx}$, and $\varepsilon_{yy}$ (Fig. 3d–f) represent the strain field distributions generated by GPA. As shown in Fig. 3d–f, near the node regions marked with the black dots, the strain varies rotationally around the nodes, which means the atomic distortion direction also varies around the nodes, thus forming web-like spiral patterns. Furthermore, at the centers of the chiral units A and B, the strain is close to zero (represented by white color in Fig. 3d–f), being consistent with the fact that the atomic displacement at the center of n-FCC and n-HCP regions is minimal. More detailed GPA results are provided in the Supplementary Information. As the number of atoms

is very large in the supercell of the ultraflat chiral BlueP, it is hard to investigate it through ab initio simulations. Nevertheless, machine learning force field approaches such as the one demonstrated recently[33] could be exploited in future studies to explore its origin at the atomic level.

**Spectroscopic characterization of ultraflat chiral BlueP**

Having determined the structure of ultraflat chiral BlueP, next, we focus on its electronic structures. Figure 4a displays the d$I$/d$V$ spectra obtained on Cu(111) without (blue line) and with ultraflat chiral BlueP (green line), respectively. On bare Cu(111), the spectrum shows the well-known contribution of surface states starting at $V_S$ (sample bias voltage) $\approx$ −0.6 V. While on the ultraflat chiral BlueP/Cu(111), it exhibits a slow upward trend when $V_S$ < +0.5 V but a fast upward trend when $V_S$ > +0.5 V. The small-energy scale STS (see Fig. 4b) indicates a metallic character for ultraflat chiral BlueP, which could be developed for the transparent electrode. To further reveal its electronic structure, we have also performed DFT calculations. As shown in Supplementary Fig. 33a, the band structure of an isolated ultraflat BlueP shows a metallic character as its lattice has a large in-plane stretch, consistent with the previous study that the bandgap decreases with the increase of the strain[9]. When considering the Cu(111) substrate (see Supplementary Fig. 33b), P and Cu hybridized bands appear near the Fermi level and it also keeps a metallic character. Also, a recent work[21] reported the fabrication of buckled BlueP with a lattice period of ~3.4 Å on Cu(111). In that work, it was found that the isolated buckled BlueP hold a large band gap. However, their experimental and DFT results show that the buckled BlueP on Cu(111) is a metallic character,

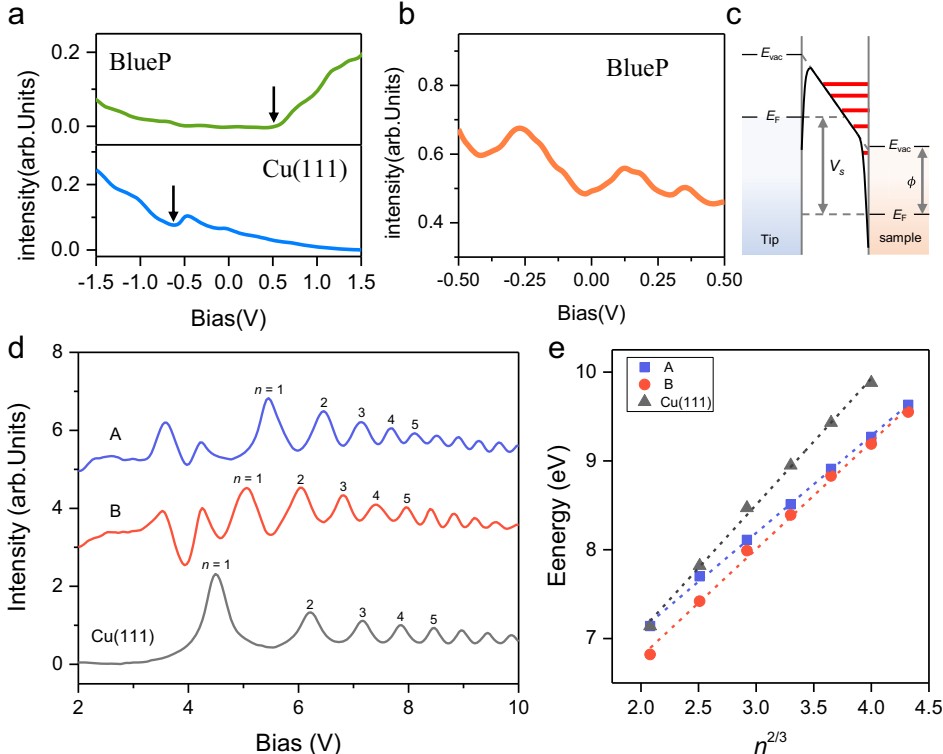

**Fig. 4 | Spectroscopic characterization of ultraflat chiral BlueP. a** Large-energy scale scanning tunneling spectroscopy (STS) acquired on Cu(111) and ultraflat chiral BlueP surface. The arrows indicate where the upward trend begins. **b** Small-energy scale STS of ultraflat chiral BlueP. The curve is an average of 64 individual spectra. **c** Schematic of the band alignment in the Fowler-Nordheim tunneling regime with Stark-shifted image-potential states (IPSs). $E_{vac}$, vacuum level; $E_F$, Fermi level; $V_s$, bias voltage; $\phi$, local work function. **d** Field-emission resonance (FER) spectroscopy obtained on chiral units A, B, and Cu(111) surface. **e** Extracted and linearly fitted IPSs peak positions from (**d**).

which was attributed to specific hybridization between the P and Cu electronic states. This is similar to the silicene on Ag(111)[34], which exhibits a strong Si-Ag hybridization. In contrast, the BlueP on Au (111) exhibits a semiconductor bandgap[35]. In order to alleviate or remove the coupling from the metal substrate and make the properties of pristine BlueP accessible, in the future, one can intercalate different atoms or molecules at the BlueP/Cu(111) interface to recover its intrinsic electronic properties, as demonstrated in graphene/metal systems[36,37].

One attractive aspect of the ultraflat BlueP on Cu(111) is the existence of chiral superstructure. What is the difference between chiral units A and B is an interesting issue. By applying a bias voltage higher than the LWF of the sample, the FER will occur when the tip Fermi level aligns to the $n$th Stark-shifted IPSs[38,39], as illustrated in Fig. 4c (red lines). Here, we performed FER spectroscopy to investigate their intrinsic difference. Figure 4d displays the series of FER spectra obtained on chiral units A, B, and Cu(111). As shown in Fig. 4d, the $n$th peaks of IPSs of chiral unit A systematically shift to higher energy compared to chiral unit B. Additionally, the $n = 1$ peaks for chiral units A and B are all higher than that on Cu(111). As the voltage of the first IPS is around the LWF[40,41], the FER spectra reveal that the LWF increases from Cu(111) to chiral unit B, and further to chiral unit A, suggesting electron transfer from the Cu substrate to BlueP. Our detailed calculations (see Supplementary Fig. 35) also reveal that electron transfer is from the substrate to BlueP, consistent with the experiment. By plotting the voltages of IPSs ($V_n$) with respect to $n^{2/3}$, as shown in Fig. 4e, the data points lie on a straight line. The LWFs extracted from the linear fitting are $\phi_A = 4.90 \pm 0.1$ eV, $\phi_B = 4.38 \pm 0.1$ eV, and $\phi_{Cu} = 4.24 \pm 0.1$ eV. The larger LWF for chiral unit A indicates a higher apparent barrier height than chiral unit B, as the apparent barrier heights correlate positively with sample LWF[42]. As reported previously[43], in Au(111) herringbone

regime, the HCP region has a higher apparent barrier height than the FCC region due to the slightly higher concentration of atoms in the HCP region. Considering the similar surface reconstruction mechanism between Au(111) herringbone and the ultraflat chiral BlueP, we suggest that chiral unit A belongs to HCP region, and chiral unit B is FCC region.

The difference of LWF between chiral units A and B can also be revealed by examining the evolution of STM images with the bias voltage (see Supplementary Fig. 17). From −2.5 V to +0.1 V, the morphologies of chiral units A and B show little difference, as they closely follow the real atomic-scale corrugation. As the bias is raised above +1.0 V, the morphologies change significantly compared with atomic corrugation (see Fig. 1g). Particularly, the apparent heights (brightness) of chiral units A and B show large difference when $V_S > +4.5$ V. And this apparent height difference between chiral units A and B show an oscillatory behavior. This peculiar behavior can be explained by spatial variations of energy shifts of the IPSs[44]. When the bias voltage reaches the first peak of the IPS in chiral unit B, unit B becomes higher (brighter) than unit A. Continuing to raise the bias voltage then reaching the first peak of IPS in unit A, the contrast would be inversed. This contrast reversal behavior at high bias voltage also confirms that the chiral units A and B have different resonance states, thus, different LWFs.

**The evolution of the phosphorene structures**

In our experiment, we observed four different phosphorene structural phases as the phosphorous coverage increases, including the chain, strip, chiral, and hexagonal phases, as shown in Fig. 5. At proper substrate temperatures (200–250 °C), a single structural phase can be obtained on the surface, and its specific structural phase depends on the coverage of phosphorus. Through tuning the substrate

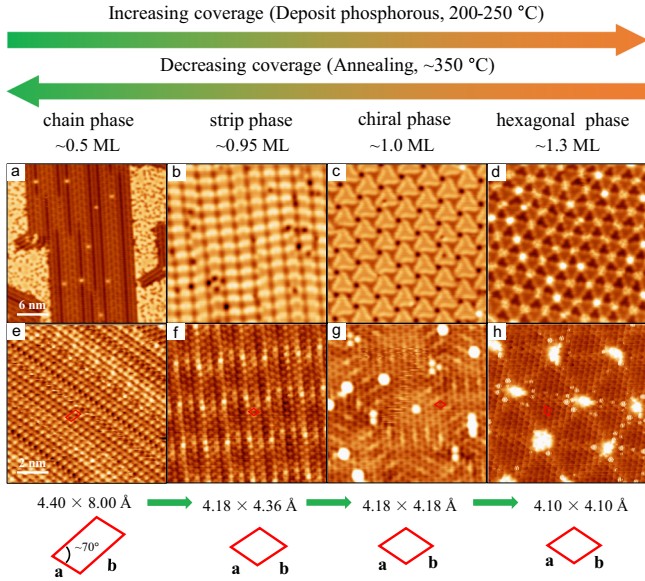

**Fig. 5 | Reversible transformations of phosphorene. a–d** STM topography images (30 × 30 nm², all the panels have the same scale bars) with different coverages of phosphorene grown on Cu(111), showing four different phosphorene structures, including three non-chiral structures (**a**, **b**, and **d**) and one chiral structure (**c**). **e–h** High-resolution STM images (10 × 10 nm², all the panels have the same scale bars) corresponding to (**a–d**). The red quadrilaterals mark its corresponding unit cell. The corresponding lattice constants are also indicated below the images.

temperature and phosphorus coverage, we can achieve the transformations from the chain to strip, to chiral, and finally to the hexagonal phase. Importantly, under proper annealing conditions (~350 °C), the phosphorous atoms can slowly desorb from the surface, decreasing coverage and achieving a completely reversible transformation (see Fig. 5). More details about the controllability of these competing phosphorene structures can be seen in Supplementary Information Note 5. On the mechanistic side, our results also indicate that the formation of atomically thin 2D phosphene sheets on the Cu (111) surface is a self-limiting process, which does not form a second layer after reaching a full layer. When the chain phase fully covers the Cu(111) surface, the deposited phosphorus atoms enter the first layer of phosphene sheets, increasing the surface density of phosphorus atoms and reducing the distance between the phosphorus atoms (as evidenced by the observation that the bond length becomes shorter as seen in the high-resolution images of Fig. 5). This evolution leads to stronger interaction between the phosphorus atoms and weaker interaction between the phosphorus and copper atoms; thus various comprehensive changes in the interfacial interaction ultimately dictate the selection of the new structural phases. The interaction strength of P-Cu atoms for different phosphorene phases is also revealed by the annealing experiments (see Supplementary Fig. 26), which shows that the chain phase is the most stable while the hexagonal phase is the most unstable, consistent with our analysis based on the bond length. Qualitatively, the stronger interaction with the substrate should induce a stronger effect on the electronic states of phosphorene by the substrate. Therefore, we suggest that the chain phase has the strongest hybridization between the P and Cu electronic states, while the hexagonal phase has the weakest hybridization. The STS results (see Supplementary Fig. 27) show that all these phosphorene phases are of a metallic character, making it difficult to quantitatively determine their specific interaction with the substrate from the STS results. To further understand how the electronic structures of different phases are affected by the substrate, further ARPES experiments and DFT calculations are needed.

## Discussion

In summary, we have obtained the large-area ultraflat chiral BlueP on Cu(111) by MBE, as large as the substrate, which is limited by the wafer size. STM results reveal that the BlueP formed on Cu(111) surface has an ultraflat honeycomb lattice and a chiral superstructure. We suggest a plausible mechanism to explain the formation of the chiral superstructure, which originates from the rotational stacking with substrate and interface strain release. The strain distribution fields of high-resolution STM images generated by GPA further confirm this explanation. However, the exact formation process of chiral superstructure may come from other mechanism, and we cannot rule out it. Moreover, through STS, we demonstrate the metallic nature of the ultraflat chiral BlueP. A systematic shift of characteristic quantum oscillations in the image-potential states is observed on adjacent chiral units, indicating higher LWFs for chiral units A than B, and both are larger than Cu(111). These results suggest that a strong interaction with the Cu(111) substrate remarkably impacts the intrinsic electronic properties of BlueP.

The demonstrated capability of controlled mass production of chiral templates may enable further developments of novel chiral functional platforms. As an example, chirality-induced spin selectivity in electron transmission has been reported experimentally in a thin chiral molecule layer grown on metal substrates[13,45], which could be used to generate highly polarized electron beams. As shown in Supplementary Figs. 17 and 18, the STM morphology and surface electronic states exhibit the periodic spiral patterns related to the chiral superstructure, indicating that the surface electronic states of BlueP are strongly modulated by the chiral superstructure as well. This character implies that different chiral phases might lead to opposite spiral potentials[46] and show different responses to the incident polarized photons or spin-polarized electrons, which can be exploited for a range of potential applications in polarization optics and spintronics. As the XPS result shows, it is vulnerable to environmental degradation (see Supplementary Fig. 2); therefore, proper encapsulation schemes such as those implemented in the cases of silicene[47,48], black phosphorus[49], and blue-phosphorene on Au[50], could be exploited in future explorations of its potential practical applications. In addition, as the surface plays an important role in many chemical processes such as adsorption, catalysis, and crystallization, developing chiral surfaces and understanding their enantioselectivity are potentially rewarding. Chiral surfaces possess enantioselective physical and chemical properties, making them suitable for many other enantioselective chemical and physical processes, including multiphase catalysis, enantioselective sensing, and crystallization[15,51]. Specifically, chiral surfaces can provide an environment for various isomeric enantioselective processes used in industries such as pharmaceuticals and agricultural chemicals. The active nature of the chiral BlueP stabilized on Cu(111) might be essential in rendering the system highly desirable for enantioselective chemical processes. Therefore, the discovered chiral BlueP could provide a platform of chiral metal surface for studying chiral catalysis, chiral recognition, and chiral separation.

## Methods
### Experiments
Sample preparation and in situ STM/STS analysis were carried out in an Omicron UHV MBE LT-STM system (base pressure: 2 × 10⁻¹⁰ mbar). The Cu(111) crystal surface was cleaned by cycles of Ar+ sputtering (1.1 keV, 10⁻⁵ mbar) and annealing (500 °C, 15 min). Phosphorus vapor was generated from InP pieces heated at ~500 °C in a standard Knudsen cell, and deposited on the substrate held at ~200 °C. The phosphorus flux was kept at ~0.1 ML per minute. After in situ STM analysis, the amount of phosphorus on some samples was measured semi-quantitatively using an ex-situ XPS facility. Our STM/STS measurements were performed on samples held at either 78 K or 4.5 K with a

PtIr tip, and the electronic properties of the tips were checked on clean Cu(111) sample. The topographic images were acquired using a constant current mode. Differential conductance d$I$/d$V$ spectra were collected through a standard lock-in technique with a modulation voltage of ~10–20 mV at 773 Hz. In the main text and the Supplementary Information, most of the STM experiments were performed at 4.5 K and the other experiments performed at liquid nitrogen temperature (78 K) were specified in the Figure caption. The XPS experiments were performed ex situ by taking the as-prepared sample out into air, and transferring it into the XPS system (AXIS SUPRA+ with a monochromatic Al Kα X-ray source, hν = 1486.6 eV).

## Computational details

The density functional theory (DFT) calculations were performed using the projector-augmented wave (PAW) method[52], implemented in the Vienna ab initio simulation package (VASP)[53], with the Perdew-Burke-Ernzerhof (PBE) exchange and correlation functional[54]. The kinetic energy cutoff of the plane-wave basis was chosen to be 400 eV, and the Brillouin zone (BZ) was sampled by Monkhorst-Pack k-mesh with a density of 2π × 0.02/Å. A correction to the total energy (DFT-D2)[55] was applied to describe the long-range van der Waals interaction between adlayer and substrate. The bulk lattice constant of Cu was scanned as 2.52 Å, while that for freestanding buckled BlueP was found as 3.28 Å. The interfacial interaction between BlueP and Cu(111) substrate was investigated by placing BlueP (1 × 1) on a 3-layer slab of Cu(111) with a surface lattice constant of 4.18 Å. The bottom layer Cu atoms were fixed as bulk geometry, and the remaining atoms were fully relaxed in all the calculations. Our calculation results revealed that due to such a large lattice mismatch, the buckled BlueP becomes flat after structure optimization. To assess the stability of the ultraflat BlueP on Cu(111) surface, we calculated the binding energy per P adatom, defined as $E_b = -(E_{total} - N_p \times E_p - E_{sub})/N_p$. Here, $E_{total}$ is the total energy of the combined system, $N_p$ is the total number of P adatoms at a given coverage, $E_p$ is the energy per P atom calculated from an isolated $P_4$ molecule, and $E_{sub}$ is the energy of the substrate. To eliminate spurious interaction between two adjacent slabs, a vacuum layer of thickness larger than 15 Å was applied. The atomic structures were fully relaxed until the force on each atom was less than 0.01 eV/Å with the bottom layer of copper atoms fixed.

# Data availability

Relevant data supporting the key findings of this study are available within the article and the Supplementary Information file. All raw data generated during the current study are available from the corresponding authors upon request.

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

## Acknowledgements

This work was supported by the National Natural Science Foundation of China (Grant Nos. 12004099, 11974323, and 12374458), the Innovation Program for Quantum Science and Technology (Grant No. 2021ZD0302800), the Plan for Leading Talent of Fundamental Research of the Central China in 2020, the Intelligence Introduction Plan of Henan Province in 2021 (No. CXJD2021008), the Anhui Initiative in Quantum Information Technologies (Grant No. AHY170000), the Strategic Priority Research Program of Chinese Academy of Sciences (Grant No. XDB0510200), the Ministry of Education, Singapore, under the Academic Research Fund Tier 1 (FY2023, Grant No. 23-0450-A0001), and the Distinguished/Outstanding Postdoctoral Fellowships of the Hefei National Laboratory for Physical Sciences at the Microscale.

## Author contributions

Z.Z., W.Z., and Y.S. conceived and designed the study. Z.Z. and W.Z. supervised the project. Y.S. performed the experiments with the help of Q.W. and Y.W., U.M. conducted the DFT calculations. Y.S., U.M., Y.J., P.C., W.Z., X.W., and Z.Z. analyzed the experimental and theoretical data. Y.S., W.Z., X.W., and Z.Z. co-wrote the paper. All authors discussed the results and commented on the paper.

## Competing interests

The authors declare no competing interests.
