## [Peer Review File · Nature Communications]

Realization of large-area ultraflat chiral blue phosphoreneREVIEWER COMMENTS

Reviewer #1 (Remarks to the Author):

The authors present a molecular beam epitaxial growth of monolayer blue phosphorene on Cu(111) and its subsequent characterization via scanning tunneling microscopy measurements. High-resolution STM images show a chiral superstructure, which may originate from the interface strain release, similar with the herringbone reconstruction on Au(111) surface.

Theoretical models and DFT calculations were also performed to better understand their observation.

This study is a nice piece of work. The STM images are of high quality, and the methodology is sound. It merits publication in the Nature Communications, once the authors address satisfactorily the following issues. A major revision is required.

Major points:

1. The authors stated the electron transfer from the substrate to the blue phosphorene layer several times. However, no such information from DFT calculation has been provided directly, e.g. Bader charge analysis.
2. No useful information can be found in Figure 1d. Although it is a large scale over 0.4 micrometers, topographic image barely can provide “atomic” information. A derivative or current image is suggested, which can show the different “morphology” instead of only “smooth” surface in present case.
3. As demonstrated in Ref. Adv. Funct. Mater, 2023, 33, 2213664, they also prepared blue phosphorene on Cu(111) but with a buckled structure (0.34 nm). In current study, the authors synthesized ultraflat phosphorene (0.42 nm). Why blue phosphorene has such tremendous difference on the same substrate? Any comment?
4. The authors have devoted efforts to illustrating the formation process of chiral superstructure. It is quite similar to the adsorption induced herringbone reconstruction on Au(111) surface. The authors are suggested to refer to a recent work from Feng Ding’s group (DOI: 10.1126/sciadv.abq2900).
5. In Figure 3b, the borderlines are illustrated to match the chiral superstructures. What is the rationale behind?
6. In Figure 4e, why the x-axis is $n2/3$?
7. In Supplementary Figure 17, there are only two images in negative bias, -0.5 and -2.5 eV. How about the -1.0 V, -1.5 V and -2.0 V? Furthermore, the positive bias can be as high as +7V. How about the negative bias below -2.5V? These images should also be provided.

Minor points:

8. It seems that the surface structure of blue phosphorene on Cu(111) depends strongly on the precursor, e.g. black P and InP. Any comment from the authors about this?
9. There are bright spots at the nodes, indicating rather different growth mode from traditional ones, like graphene on Cu(111), on which there is only moiré spots. Any comments?
10. What is the ratio of R and L phases within the blue phosphorene? Which one is more energetically favorable?

Reviewer #2 (Remarks to the Author):

The paper reports on the growth of P/Cu(111) which induces the formation of blue phosphorene.

The authors did not report significant results to justify their publication in Nature communications since the most important results were already reported in ref 20 cited in the manuscript.

Below my comments:

-The title of the paper is misleading. The authors used the word wafer scale.

We can think that the phosphorene was grown on a Cu foil like in the case of Graphene while the substrate used in this study was only a Crystal substrate with probably few mm of size.

-It was reported in ref 20 that the deposition of 1 ML of P on Cu(111) induces a layer of phosphorene and beyond 1 ML a formation of self-assembled nanodots were obtained. The authors did not discuss this point in their paper. Probably this can explain what they call chiral phase.

The authors present (Fig. S17) the effect the bias voltage on the STM images showing the changes of the density of state versus the bias voltage. Probably what they observe is only the results of the bias voltage. Further investigations are necessary to conclude.

-The authors claim that the phosphorene is ultraflat like graphene. If this is the case, they should observe by STM the first nearest P atoms. In all their images, only the second nearest P atoms were observed. This is a signature of buckling.

In addition, they gave a unit cell distance of 4.41 Å, which gives a P-P first nearest atoms distance of 2.54 Å. This value is very big compared to the expected one (2.25 Å).

This is probably because the authors used Cu(111)-($\sqrt{3} \times \sqrt{3}$)R30° superstructure instead of Cu(111)-0.8($\sqrt{3} \times \sqrt{3}$)R30° reported in ref 20.

-I don't understand why the authors performed ex-situ XPS measurements on phosphorene. It is well known that phosphorene is not stable in air. This doesn't add any added value to the manuscript. In-situ measurements are necessary to explain the different structures.

Reviewer #3 (Remarks to the Author):

The manuscript by Song et al. reports on the macroscale area epitaxy of ultra flat blue-phosphorene on (111) terminated Cu substrate with asymmetric bimodal structure displaying a chiral character. The emergence of the ultra-flat and chiral character as salient features of the so developed epitaxy are explained in terms of the subtle interaction of the P atom arrangement with the substrate. Based on scanning tunnelling spectroscopy, the so grown blue phosphorene exhibits a local metallic character. Structural variations in the phosphorene structure are also investigated as a function of the growth

condition in order to disclose an exhaustive number of achievable motifs. Getting through the scalability of 2D phosphorus is still challenging and deserves consideration, but I have some major (1-3) and technical (4-9) concerns on the present submission as follows that should be addressed to have the manuscript reconsidered.

1) For the sake of transparency, I already happened to review the present manuscript for another journal with more stringent standards of publication. After that stage, I realise the present submission includes some little changes consisting of the inclusion of Figure 5 as an overview of the possible phosphorene structures, and of Ref. 20 by Kaddar et al. Ref. 20 firstly reported on the large scale epitaxy of blue-phosphorene on Cu(111) methodologically in complete analogy with the present work. This is a strong limitation to the prior novelty of the work if the Authors aim at a high impact publication. However, the progress on the phosphorene growth are demanding in consideration of the fact that black phosphorus is hardly scalable (hardly but not impossibly, as reported in Nature Materials 20, 1203 (2021)) and the present work is step beyond the current knowledge on the matter. However, while in Ref. 20 phosphorene is observed to be a Dirac semimetal, the present manuscript reports on the metallic character of the blue-phosphorene on Cu(111) as derived from local scanning tunnelling spectroscopy. To avoid a conflict of statements, this discrepancy shall be mandatorily discussed and elaborated in a revised version of the present submission. In the end, it should be clear in what terms the cases differentiate and what is the true electronic character(s) of the epitaxially grown phosphorene on Cu(111).

2) Telling the true electronic character is also critical to assess the impact of the blue-phosphorene itself (and even its name, does it make sense to name it blue-phosphorene if it is metallic? Perhaps chiral / epitaxial phosphorene is more appropriate) in the full framework of the Xenes. Indeed, the evidence of a purely metallic nature limits the significance of the materials itself with respect to the wealth of more intriguing and technology relevant configurations that can be found in the Xenes (see for instance the topological character in stanene, bismuthene, the semi metallic character in silicene, etc.). In origin, blue phosphorus, namely rhombohedral phosphorus, is expected to be semiconducting even at its 2D state (Phys. Rev. Lett. 112, 176802). It is then plausible to assign its metallic character to the interaction with its substrate as happens with other Xenes like silicene (see Phys Rev B 92, 165427 (2015)). It would be interesting to assess to what extent the metallic character is induced by proximity with the metal substrate and how this proximity can be alleviated or removed.

3) The evidence of a chiral structure in the so-grown phosphorene is intriguing but at the present there is no implication of chirality in any potentially exploitable physical property (for instance, optical dichroism, or so). This makes it an interesting result of the 2d growth with a short momentary impact. I recommend the Authors to go through the potential of the chiral structure in terms of possible implications or at least, discuss what they prospect to be gained by taking benefit from the chiral phosphorene.

4) Fig. 1 is not satisfactory in giving an atomically resolved structure of the blue-phosphorene cell to be more closely matched with that debated in the case of the epitaxy on Au(111) where different models - Au-P framework [J. Phys. Chem. C 2020, 124, 2024] and adatom-rich structure [Nano Lett. 2016, 16, 4903] - were proposed to rationalise the local STM topography. Data on the atomic structure should more extensively shown;

5) what is the physical meaning of the bright protrusions in Fig. 1g at the vertex of the spiral triangles? Even in this case, an atomistic picture of the phosphorene structure is due for the sake of a more exhaustive understanding.

6) no insight is given through the environmental stability of the so-grown phosphorene while it is known that buckled blue-phosphorene as well as puckered black phosphorene are vulnerable to environmental degradation [Nanoscale 2019,11, 18232 and Nano Lett. 2014, 14, 6964, respectively] and its encapsulation is compulsory for every deployment in application.

7) while the origin of the chiral structure is attributed to the differential match with the substrate, it is not clear how the chirality of phosphorene motifs and hence, its twisted orientation with the substrate can be effectively controlled by a proper surface engineering or so. If this aspect is not clarified, since the epitaxy is not a controllable top-down scheme, I'm afraid that chiral structures as observed could turn out to be randomly emerging with no possible control over it thus limiting its applicability.

8) to what extent the chiral motifs in blue-phosphorene are statistically supported? And to what extent is the epitaxy of epitaxial phosphorene affected by growth parameter variability? For instance, what is the influence of the temperature in the formation of ultra-flat phosphorene and hence, in the emergence of the observed chiral motifs? Aiming at the dissemination through Science should entail a statistically extensive examination of the case in all the configurations to get rid of possible singular or local events. In the present case, the manuscript appears to be quite short in the extensive consideration of the experimental case which is reduced to a single, though interesting, configuration.

9) While commenting on Fig. 5 the Authors state that "various comprehensive changes in the interfacial interaction ultimately dictate the selection of the new structural phases". It should be expected to understand how the variable interaction with the substrate affects the electronic character of the phosphorene. I recommend the Authors to elaborate more on this aspect.

Responses to Reviewers' Reports

(MS # NCOMMS-23-33683A by Yeheng Song et al.)

The detailed responses are listed below, and the manuscript has been revised accordingly. With these responses and revisions, we hope that this paper is now acceptable for publication in Nature Communications.

Detailed responses to Reviewer 1

Generic Comments: *The authors present a molecular beam epitaxial growth of monolayer blue phosphorene on Cu(111) and its subsequent characterization via scanning tunneling microscopy measurements. High-resolution STM images show a chiral superstructure, which may originate from the interface strain release, similar with the herringbone reconstruction on Au(111) surface.*

Theoretical models and DFT calculations were also performed to better understand their observation.

This study is a nice piece of work. The STM images are of high quality, and the methodology is sound. It merits publication in the Nature Communications, once the authors address satisfactorily the following issues. A major revision is required.

Responses: We thank the reviewer again for the succinct summary of the main findings and for recommending acceptance of this work in Nature Communications after proper revision. In the following, we reply to the various comments and suggestions point-by-point.

Major points:

Comment 1: *The authors stated the electron transfer from the substrate to the blue phosphorene layer several times. However, no such information from DFT calculation has been provided directly, e.g. Bader charge analysis.*

Responses: Following the reviewer's suggestion, we have calculated the charge density redistribution of the ultraflat BlueP on Cu (111), as shown in Fig. R1. The charge density difference $\Delta\rho_c$ is defined as the plane-averaged charge difference along the c -axis between the BlueP/Cu(111) system and the sum of the isolated BlueP monolayer and Cu(111) substrate. It can be seen that when the BlueP and Cu layers merge into the BlueP/Cu(111), an electron-depletion region is formed near the top most layer of Cu, and this amount of electron charge is transferred to the interfacial region. Taking a physically intuitive reference plane where the total charge is a minimum (highlighted by the black dash line in Fig. R1b), the amount of electron transfer can be obtained by integrating $\Delta\rho_c$ over the c -axis within the BlueP region. The net charge transfer thus calculated is 0.018 e/f.u. from the Cu(111) substrate to BlueP. For comparison, we have also calculated the charge transfer through the Bader charge analysis. Our calculated results reveal that the charge transfer from the Bader charge analysis (0.09 e/f.u.) and through integrating the charge density difference distribution (0.018 e/f.u.) show the same direction of charge transfer, but the former approach could overestimate its magnitude because the Bader analysis relies on the value of peaked charge. Such a significant charge transfer and a strong binding energy (0.56 eV/P) explain the strong interaction between the P atoms and Cu(111), resulting in the formation of the ultraflat BlueP.

In response to the Reviewer's comment, we have added Fig. R1 to Supplementary information as new Fig. S35. On page 6 in the main text, we have added the sentence **“Our detailed calculations (see Supplementary Fig. 35) also reveal that electron transfer is from the substrate to BlueP, consistent with the experiment.”**

In addition, we have added the following section about the calculation details of the charge transfer to the Supplementary Information.

“Supplementary Note 7. Calculation of charge density distribution of the ultraflat BlueP on Cu (111).

To quantify the amount of charge transfer, we have calculated the charge density distribution of the ultraflat BlueP on Cu (111), as shown in Fig. S35. The charge density difference $\Delta\rho_c$ is defined as the plane-averaged charge difference along the c -axis between the BlueP/Cu(111) system and the sum of the isolated BlueP monolayer and Cu(111) substrate. It can be seen that when the BlueP and Cu layers merge into BlueP/Cu(111), an electron-depletion region is formed near the top most layer of Cu, and this amount of electron charge is transferred to the interfacial region. Taking a physically intuitive reference plane where the total charge is a minimum (highlighted by the black dash line in Fig. R1b), the amount of electron transfer can be obtained by integrating $\Delta\rho_c$ over the

c-axis within the BlueP region. The net charge transfer is calculated to be 0.018 e/f.u. from the Cu(111) substrate to BlueP. For comparison, we have also calculated the charge transfer through the Bader charge analysis. Our calculated results reveal that the charge transfer from the Bader charge analysis (0.09 e/f.u.) and through integrating the charge density difference distribution (0.018 e/f.u.) show the same direction of charge transfer, but the former approach could overestimate its magnitude because the Bader analysis relies on the value of peaked charge. Such a significant charge transfer and a strong binding energy (0.56 eV/P) explain the strong interaction between the P atoms and Cu(111), resulting in the formation of ultraflat BlueP.”

Figure. R1 a, Charge density difference of the ultra-flat BlueP on Cu (111). The cyan (yellow) zone gains (losses) charge with isosurface value of $\pm 0.004|e|/\text{bohr}^3$. b,c, Total charge and charge density difference distributions for the ultraflat BlueP on Cu(111).

Comment 2: *No useful information can be found in Figure 1d. Although it is a large scale over 0.4 micrometers, topographic image barely can provide “atomic” information. A derivative or current image is suggested, which can show the different “morphology” instead of only “smooth” surface in present case.*

Responses: It is a good suggestion that the derivative image might help to enhance the detailed morphology. Figure R2b shows the derivative STM image of Fig. 1d. Yet due to the very large scanning area, it is hard to distinguish the detailed chiral superstructure from this derivative STM image. Nevertheless, from the large-area derivative STM image (see Fig. R2b), one can distinguish that there are two types of domains with different morphology on the BlueP surface (as marked by D1 and D2), which belong to two different chiral phases. Additionally, as shown in Fig. S19, the different phases (R or L phase) show different moiré patterns in large-area STM images when scanning with low pixels, and the underlying principle for such a higher resolution can be seen in the Figure Caption of Fig. S19.

In response to the reviewer's comment, Figure 1d has been replaced by Fig. R2a, and Fig. R2b-d has been added to Supplementary information as new Fig. S20. On page 3 in the main text, the sentence “**Figure 1d is the large-area STM image after phosphorus deposition, where the phosphorene sheets completely cover all the terraces.**” has been revised to “**Figure 1d is the large-area STM image after phosphorus deposition, where the phosphorene sheets completely cover all the terraces, and there exist two different domain areas (see Supplementary Fig. 20).**”.

Figure. R2 a,b, Large-area STM topography image and its corresponding derivative STM image of ~1 ML ultraflat chiral BlueP, showing two types of domain, D1 domain with an obvious period structure and D2 domain with a relatively flat morphology. **c**, The enlarged derivative STM image in the D2 domain. **d**, The enlarged derivative STM image in the D1 domain.

Comment 3: *As demonstrated in Ref. Adv. Funct. Mater, 2023, 33, 2213664, they also prepared blue phosphorene on Cu(111) but with a buckled structure (0.34 nm). In current study, the authors synthesized ultraflat phosphorene (0.42 nm). Why blue phosphorene has such tremendous difference on the same substrate? Any comment?*

Responses: We are grateful to Reviewer 1 for this critical and intriguing question. We noted that the buckled BlueP [Adv. Funct. Mater, 33, 2213664 (2023)] grown on Cu(111) used a higher growth temperature ($\sim 260^\circ\text{C}$) and black phosphorus as the precursor. As a comparison, in our experiments, we use a relatively low growth temperature ($200\text{-}250^\circ\text{C}$) and InP as the precursor. To investigate why BlueP has such a tremendous difference on the same substrate, we have performed additional supplementary experiments. Figure R3 shows the STM topographic images of the phosphorene structures grown at $\sim 260^\circ\text{C}$ [the growth temperature used in Ref. Adv. Funct. Mater, 33, 2213664 (2023)] but still using InP as the precursor, which also shows the same phosphorene structures (chain, strip, and chiral phases) as the results of Fig. 5 in the manuscript (growth temperature: $200\text{-}250^\circ\text{C}$). It is known that below 800°C , when using black phosphorus as the precursor, phosphorus vapor mainly consists of P_4 molecules [Nano Lett. 16, 4903-4908 (2016)]. However, with InP in the evaporator, the phosphorus vapor mainly consists of P_2 molecules [Semiconductor Science and Technology 24, 055014 (2009)]. Therefore, we speculate that the different phosphorus precursors might result in different phosphorene structures on Cu(111). However, we cannot rule out the influence of the growth temperature. Due to certain deviations between the measured substrate temperature and the actual substrate temperature, which is different for different MBE systems, the actual temperature used in our experiment might not be the same as the work of Kaddar et al. [Ref. Adv. Funct. Mater, 33, 2213664 (2023)]. Therefore, to further understand the influence of the phosphorus precursor and growth temperature on phosphorene structures grown on Cu(111), more control and comparative experiments are needed.

In response to the Reviewer's comments, Figure R3 has been added to the Supplementary Information as new Fig. S22, and the following sentences have been added on page 8 in the Supplementary Information.

“In addition, we noted that the recent work of Kaddar et al. [Adv. Funct. Mater, 33, 2213664 (2023)] reported the fabrication of a buckled BlueP with a lattice period of 3.4 \AA on Cu(111). In their work, they used a higher growth temperature ($\sim 260^\circ\text{C}$) and black phosphorus as the precursor. As a comparison, in our experiments, we use a relatively low growth temperature ($200\text{-}250^\circ\text{C}$) and InP as the precursor. To investigate why BlueP has such a tremendous difference on the same substrate, we have tried to grow phosphorene at a higher growth temperature. Figure S22 shows the STM topographic images of the phosphorene structures grown at $\sim 260^\circ\text{C}$ (the growth temperature used in Ref. Adv. Funct. Mater, 33, 2213664 (2023)) and still uses InP as the precursor, which

also shows the same phosphorene structures (chain, strip, and chiral phases) as the results of Fig. 5 (substrate temperature: 200-250°C). It is known that at temperature below 800 °C, when using black phosphorus as the precursor, phosphorus vapor mainly consists of P₄ molecules [Nano Lett. 16, 4903-4908 (2016)]. However, with InP in the evaporator, the phosphorus vapor mainly consists of P₂ molecules [Semiconductor Science and Technology 24, 055014 (2009)]. Therefore, we speculate that the different phosphorus precursors might result in different phosphorene structures on Cu(111). However, we cannot rule out the influence of the growth temperature. Due to a certain deviation between the measured substrate temperature and the actual substrate temperature, which is different for different MBE systems, the actual temperature used in our experiment might not be the same as the work Kaddar et al. Therefore, to further understand the influence of the phosphorus precursor and growth temperature on phosphorene structures grown on Cu(111), more control and comparative experiments are needed.”

Figure. R3 STM topographic images after growth of ~0.85 ML, 0.9 ML, and ~1 ML phosphorous on Cu(111), in which the substrate is kept at ~260 °C (the growth temperature used in Ref. Adv. Funct. Mater, 33, 2213664 (2023)) and uses InP as the precursor.

Comment 4: *The authors have devoted efforts to illustrating the formation process of chiral superstructure. It is quite similar to the adsorption induced herringbone reconstruction on Au(111) surface. The authors are suggested to refer to a recent work from Feng Ding’s group (DOI: 10.1126/sciadv.abq2900).*

Responses: As we suggested in the manuscript, the chiral superstructure can be attributed to the contrasting rotational stacking with respect to substrate and the interface strain release generated by the mismatch of the heteroepitaxial lattice. It’s quite similar to the adsorption induced herringbone reconstruction on the Au(111) surface. As these systems possess very large number of atoms, they cannot be addressed by ab initio simulations at present. In a recent work [Sci. Adv. 8, eabq2900 (2022)], Ding et al. explored the origin of the herringbone reconstruction on Au(111) surface at the atomic level with the help of the machine learning

force field with high accuracy. Here, we have cited this work as Ref. 33. hoping that, one can refer to their work to investigate the origin of ultraflat chiral BlueP at the atomic scale.

In response to the Reviewer's comment, on page 5 in the main text, we have added the sentence **“As the number of atoms is very large in the supercell of the ultraflat chiral BlueP, it is hard to investigate it through ab initio simulations. Nevertheless, machine learning force field approaches such as the one demonstrated recently [Ref. 33] could be exploited in future studies to explore its origin at the atomic level.”**.

Comment 5: *In Figure 3b, the borderlines are illustrated to match the chiral superstructures. What is the rationale behind?*

Responses: As we proposed, the chiral superstructure results from the rotational stacking with respect to the substrate and interface strain release. In a previous work, the Cu(111)/Ni(111) interface (a similar system to our experiment, Ref. Sci. Rep. 3, 1-7 (2013)) can form different superstructures (chiral and hexagonal) as the lattice mismatch and twisted angle vary. In Figure 3b, the borderline is a schematic diagram, indicating that under appropriate lattice mismatch and twisted angle, a spiral borderline will be formed to separate the FCC and HCP regions in the supercell. These spiral borderlines correspond to the edges of the spiral triangular structure of the chiral superstructure observed in our experiments. However, if the lattice mismatch and twisted angle vary, it could form other structures, such as the hexagonal phase with a straight borderline, as shown in Fig. 5h. The observed two phases (chiral and hexagonal) are consistent with previous theory [Sci. Rep. 3, 1-7 (2013)], further supporting the mechanism we proposed.

In response to the Reviewer's comment, on page 5 in the main text, the sentences **“To minimize the surface energy, the structure relaxes via atomic displacement, the near-FCC and near-HCP regions transform into FCC and HCP regions, called chiral units A and B, respectively, separating by a web-like network of borderlines.”** have been revised to **“To minimize the surface energy, the structure would relax via atomic displacement. Under appropriate lattice mismatch and twisted angle, the near-FCC and near-HCP regions transform into FCC and HCP regions (called chiral units A and B, respectively), separated by a spiral network of borderlines.”**.

Comment 6: *In Figure 4e, why the x-axis is $n2/3$?*

Responses: From a quasi-classical approximation, when the sample bias voltages align with one of these shifted image potential states (IPSs), resonant tunneling occurs as [Phys. B 291, 246-255 (2000) and Nat. Mater. 21, 35–40 (2021)]

$$eV_n = \phi + \left(\frac{3n\pi\hbar eE}{2\sqrt{2}m} \right)^{2/3}, \quad (1)$$

where V_n is the sample voltage of the n th resonant, ϕ is the local work function, and E is the electric field. By plotting V_n of the n th IPS with respect to $n^{2/3}$ in Figure 4e, the data points lie along straight lines in agreement with Eq. (1). The local work functions are therefore determined by the y-axis intercepts of linear fits to $V_n - n^{2/3}$ data points. For better accuracy, only $n \geq 3$ IPSs are used for fitting. If we plot the V_n of the n th IPS with respect to n (as shown in Figure R4), the data points will not lie along straight lines, the work functions also can be obtained through fitting, but not as simple as in the linear fitting.

In response to the Reviewer's comment, we have added Fig. R4 to the Supplementary Information as the new Fig. S28.

Figure. R4 Extracted image-potential state peak positions from Fig. 4d, which plot the V_n of the n th IPS with respect to n .

Comment 7: *In Supplementary Figure 17, there are only two images in negative bias, -0.5 and -2.5 eV. How about the -1.0 V, -1.5 V and -2.0 V? Furthermore, the positive bias can be as high as +7V. How about the negative bias below -2.5V? These images should also be provided.*

Responses: Figure R5 shows the updated Fig. S17 in Supplementary Information, which now contains the new STM images taken at -1.0 V, -1.5 V, -2.0 V, -3.0V, and -3.5V. As one can see, below 1.0 V, the morphologies show little difference. We also tried to scan the STM image below -3.5 V, but we found the STM tip cannot be stable when the tip bias is below -3.5 V.

Figure. R5 STM morphologies of ultraflat chiral BlueP on Cu(111) with vary bias voltage. Bellow 1.0 V, the morphologies show little difference. However, above 1.0 V, the morphologies show rich forms. Specially at high scanning bias, the brightness (apparent height) of triangle units shows oscillatory variation.

Minor points:

Comment 8: *It seems that the surface structure of blue phosphorene on Cu(111) depends strongly on the precursor, e.g. black P and InP. Any comment from the authors about this?*

Responses: This comment has been addressed in our response to Comment 3 earlier.

Comment 9: *There are bright spots at the nodes, indicating rather different growth mode from traditional ones, like graphene on Cu(111), on which there is only moiré spots. Any comments?*

Responses: As shown in Fig. 1g, the bright protrusions are located at the positions of the nodes. The atomic resolution STM image (see Fig. R6) shows that bright protrusions are composed of three P atoms (a phosphorous trimer). From the line-scan profile across the protrusions (see Fig. S4), the measured height of the protrusion is $\sim 0.8 \text{ \AA}$, indicating that the phosphorous trimers are located on top of the phosphorene layer. We note that most of the bright protrusions

are phosphorous trimers, and only a small portion of them are dimers or tetramers (see Fig. 2). As proposed in the schematic diagrams of Figs. 3a and 3b, at the node, the first-layer atom is located at the on-top site, which is not a stable site. Therefore, the atoms in the nodal area relax via atomic displacement to minimize the total energy. As shown in Fig. R6c, there exist obvious misfit dislocations at the nodal area due to the atomic displacement, as guided by the black lines. Therefore, this chiral pattern is very different from the moiré spots reported in buckled BlueP [Adv. Funct. Mater, 33, 2213664 (2023)] grown on the Cu(111) substrate.

In response to the Reviewer's comment, Figure S13 has been replaced by Fig. R6 in the Supplementary Information.

On page 4 in the main text, the sentence **“The atomic resolution STM image (see Supplementary Fig. 13) and line-scan profile (see Supplementary Fig. 4) show that these bright protrusions at the nodes are composed of three P atoms (a P trimer) located on the phosphorene layer.”** has been added.

On page 3 in the Supplementary Information, we have added the sentences **“We note that there are bright protrusions at the nodes. The atomic resolution STM image (see Fig. S13) shows that these bright protrusions are composed of three P atoms (P trimer). From the line-scan profile across the protrusions (see Fig. S4), the measured height of the protrusion is ~ 0.8 Å, indicating that the phosphorous trimers are located on top of the phosphorene layer. Most of the bright protrusions are phosphorous trimers, and only a small portion of them are dimers or tetramers (see Fig. 2). As proposed in the schematic diagrams of Figs. 3a and 3b, at the node, the first-layer atom is located at the on-top site, which is not a stable site. Therefore, the atoms in the nodal area relax via atomic displacement to minimize the total energy. As shown in Fig. S13c, there exist obvious misfit dislocations at the nodal area due to the atomic displacement, as guided by the black lines. Therefore, this chiral pattern is very different from the moiré spots as reported in the buckled BlueP [Adv. Funct. Mater, 33, 2213664 (2023)] grown on the Cu(111) substrate.”**

Figure. R6 a,b, Atomic resolution STM images (10×10 nm², 78K) of ultraflat chiral BlueP,

with the green dots in **(b)** marking the P atoms at the nodes. **b**, Bragg-peaks filtered image corresponding to **(a)**, which can enhance the clarity of the lattice. The green dots label the P atoms at the nodes, and the yellow dots mark the positions of ultraflat chiral BlueP lattices. There exist obvious misfit dislocations as guided by the black lines.

Comment 10: *What is the ratio of R and L phases within the blue phosphorene? Which one is more energetically favorable?*

Responses: As proposed in our manuscript, the R and L phases are formed by two opposite stacking angles with respect to the substrate (see Fig. S7 and S8). Those two stacking ways are energetically degenerate. Therefore, the ratio of R and L phases within the BlueP should be 1:1 on the surface. As shown in Figure S19, it can be observed that the R and L phases have different moiré patterns in large-area STM images with low pixels (the detailed principle can be seen in the Figure Caption of Figure S19). From our experimental result (as shown in Figure R7), our rough statistical analysis yielded an area ratio of about 1:1 for the R and L phases, which is consistent with our proposal.

In response to the Reviewer's comment, we have added Fig. R8 to the Supplementary Information as the new Fig. S21. On page 3 in the Supplementary Information, the sentence “**From our observations (see Fig. S21), the ratio of left and right-handed chiral phase is close to 1:1, consistent with our proposal that the chiral phases are energetically degenerate.**”. has been added.

Figure. R7 Large-area STM images of ultraflat chiral BlueP covered on Cu(111), with the different chiral domain areas showing different morphologies as marked in the images. The ratio of the two domains is close to 1:1.

Detailed responses to Reviewer 2

Generic Comments: *The paper reports on the growth of P/Cu(111) which induces the formation of blue phosphorene.*

The authors did not report significant results to justify their publication in Nature communications since the most important results were already reported in ref 20 cited in the manuscript.

Responses: As briefed at the beginning of this response, the appearance of the recent work of buckled BlueP on Cu(111) [Kaddar et al., Adv. Funct. Mater. 33, 2213664 (2023), published after our initial submission to a different journal] might have weakened the originality of our work at first glance, but at a closer look, the structure formed was distinctly different from what we have achieved here. The buckled structure stabilized on Cu(111) mainly confirmed a prediction by some of the team members in 2017 [Ref. 17, Zeng et al., Phys. Rev. Lett. 118, 046101 (2017)], while here, we show for the first time that the ultraflat BlueP, an analog of graphene, can be stabilized on Cu(111) with novel chirality, and our central findings are distinctly different from all previous reports such systems. In addition, we have achieved a reversible transformation between the chiral and achiral phases of ultraflat BlueP, which reveal a distinctive growth mode for phosphorene structures.

Aside from the novel findings on growth, we would like to add that the present work on ultraflat chiral BlueP may have far-reaching fundamental and practical implications. At a fundamental level, we believe the reversible transformation between the chiral and achiral phases of ultraflat BlueP offers new insights into the ever-inspiring secrets of emergent chirality in nature. At a practical level, the demonstrated capability of controlled mass production of chiral templates may enable further developments of novel chiral functional platforms. As an example, chirality-induced spin selectivity in electron transmission has been reported experimentally in a thin chiral molecules layer grown on metal substrates [Science 283, 814-816 (1999), Science 331, 894-897 (2011)], which could be used to generate highly polarized electron beams. As shown in Fig. S17 and S18, the STM morphology and surface electronic states exhibit a periodic spiral pattern related to the chiral superstructure, indicating that the surface electronic states of BlueP are strongly modulated by the chiral superstructure as well. This character implies that different chiral phases might lead to opposite spiral potentials [Adv. Mater 34, 2106629 (2022)] and show different responses to the incident polarized photons or spin-polarized electrons, which can be exploited for a range of potential applications in polarization optics and spintronics. In addition, as the surface plays an important role in many chemical processes, such as adsorption, catalysis, and crystallization, developing chiral surfaces and understanding their enantioselectivity potentially rewarding. Chiral surfaces possess

enantioselective physical and chemical properties, making them suitable for many other enantioselective chemical and physical processes, including multiphase catalysis, enantioselective sensing, and crystallization [Nat Mater 19, 939-945 (2020), ACS Nano 4, 5-10 (2010), Catalysis Letters 145, 220-232 (2014)]. Specifically, chiral surfaces can provide an environment for various isomeric enantioselective processes used in industries such as pharmaceuticals and agricultural chemicals. The active nature of the chiral BlueP stabilized on Cu(111) might be essential in rendering the system highly desirable chemical functionalities and its chiral surfaces can serve as media for enantioselective chemical processes, which benefits from their enantiospecific interactions with chiral adsorbates. Therefore, the newly discovered chiral BlueP could provide a new platform of chiral metal surface for studying chiral catalysis, chiral recognition, and separation.

Collectedly, we believe the present manuscript meet the high standard of absolute novelty and significance as the Nature Communications readership expects.

Comment 1: *The title of the paper is misleading. The authors used the word wafer scale. We can think that the phosphorene was grown on a Cu foil like in the case of Graphene while the substrate used in this study was only a Crystal substrate with probably few mm of size.*

Responses: We agree that the title of the paper might, to some extent, mislead the readers. In our experiments, limited by the sample holder of our MBE and STM systems, the typical size of the sample is smaller than 10 mm×10 mm. In the manuscript, the Cu(111) crystal used to grow phosphorene is 9 mm in diameter. We believe using a wafer-scale MBE system can also achieve the wafer-scale ultraflat chiral BlueP (such as 2-inch, 4-inch size) on the wafer-scale Cu(111) crystal substrate. Here, to be more rigorous, we have replaced the word “**wafer-scale**” with “**large-area**” in the title. In the manuscript, we also replaced “**wafer-scale**” with “**large-area**”.

Comment 2: *It was reported in ref 20 that the deposition of 1 ML of P on Cu(111) induces a layer of phosphorene and beyond 1 ML a formation of self-assembled nanodots were obtained. The authors did not discuss this point in their paper. Probably this can explain what they call chiral phase.*

The authors present (Fig. S17) the effect the bias voltage on the STM images showing the changes of the density of state versus the bias voltage. Probably what they observe is only the results of the bias voltage. Further investigations are necessary to conclude.

Responses: We noted that the recent work of Kaddar et al. [Ref. 20] reported the fabrication of a buckled BlueP with a lattice period of 3.4 Å on Cu(111). In their work, they used a higher growth temperature (~260 °C) and black phosphorus as the precursor. As a comparison, in our experiments, we use a relatively low growth temperature (200-250°C) and InP as the precursor.

To investigate why BlueP has such a tremendous difference on the same substrate, we have performed additional supplementary experiments. Figure R3 shows the STM topographic images of the phosphorene structures grown at ~ 260 °C (the growth temperature used in Ref. 20) and but still uses InP as the precursor, which also shows the same phosphorene structures (chain, strip, and chiral phases) as the results of Fig. 5 in the manuscript (substrate temperature: 200-250°C). It is known that at a temperature below 800 °C, when using black phosphorus as the precursor, phosphorus vapor mainly consists of P_4 molecules [Nano Lett. 16, 4903-4908 (2016)]. However, for the InP precursor, the phosphorus vapor mainly consists of P_2 molecules [Semiconductor Science and Technology 24, 055014 (2009)]. Therefore, we speculate that the different phosphorus precursors might result in different phosphorene structures on Cu(111). However, we cannot rule out the influence of the growth temperature. Due to certain deviations between the measured substrate temperature and the actual substrate temperature, which is different for different MBE systems, the actual temperature used in our experiment might not be the same as the work of Kaddar et al. [Ref. 20]. Therefore, to further understand the influence of the phosphorus precursor and growth temperature on phosphorene structures grown on Cu(111), more control and comparative experiments are needed.

For STM images, the different bias voltage windows capture different integrated charge densities. When the bias voltage $V_s > +4.5$ V, the apparent height of chiral units A and B shows a large difference, and this apparent height difference shows an oscillatory behavior. This peculiar behavior can be explained by spatial variations of the energy shifts of the image-potential states, which has been reported in previous experiment [Nano Lett 12, 5821-5828 (2012)]. When the bias voltage $+1 \text{ V} < V_s < +4.5 \text{ V}$, the morphologies change significantly compared with atomic corrugation. In the dI/dV mapping (see Fig. S18), the local density of states also shows the periodic structure related to the chiral structure, indicating the surface density of states is strongly modulated by the chiral superstructure within this bias voltage range. When the bias voltage is below +1V, the morphologies of chiral units A and B show little difference, as they closely follow the real atomic-scale corrugation. The dI/dV mappings show a consistent result, in which the density of states shows a random distribution and is not modulated by the chiral superstructure. Though the morphologies change significantly when $V_s > +1\text{V}$, it always keeps a chiral pattern (see Fig. S17), indicating that the chiral superstructure is bias voltage independent.

In response to the Reviewer's comments, Figure R3 has been add to the Supplementary Information as new Fig. S22, and the following sentences have been added to page 8 in Supplementary Information.

“In addition, we noted that the recent work of Kaddar et al. [Ref. 20] reported the fabrication of a buckled BlueP with a lattice period of 3.4 Å on Cu(111). In their work, they used a higher growth temperature (~ 260 °C) and black phosphorus as the precursor.

As a comparison, in our experiments, we use a relatively low growth temperature (200-250°C) and InP as the precursor. To investigate why BlueP has such a tremendous difference on the same substrate, we have tried to grow phosphorene at a higher growth temperature. Figure S22 shows the STM topographic images of the phosphorene structures grown at ~260 °C (the growth temperature used in Ref. 20) and still uses the InP as the precursor, which also shows the same phosphorene structures (chain, strip, and chiral phases) as the results of Fig. 5 in the manuscript (substrate temperature: 200-250°C). It is known that at temperature below 800 °C, when using black phosphorus as the precursor, phosphorus vapor mainly consists of P₄ molecules [Nano Lett. 16, 4903-4908 (2016)]. However, with the InP in the evaporator, the phosphorus vapor mainly consists of P₂ molecules [Semiconductor Science and Technology 24, 055014 (2009)]. Therefore, we speculate that the different phosphorus precursors might result in different phosphorene structures on Cu(111). However, we cannot rule out the influence of the growth temperature. Due to certain deviations between the measured substrate temperature and the actual substrate temperature, which is different for different MBE systems, the actual temperature used in our experiment is might not the same as the work [Ref. 20] of Kaddar et al. Therefore, to further understand the influence of the phosphorus precursor and growth temperature on phosphorene structures grown on Cu(111), more control and comparative experiments are needed.”

Figure. R3 (reproduced) STM topographic images after growth of ~0.85 ML, 0.9 ML, and ~1 ML phosphorous on Cu(111), in which the substrate is kept at ~260 °C (the growth temperature used in Ref. Adv. Funct. Mater, 33, 2213664 (2023)) and uses InP as the precursor.

Comment 3: *The authors claim that the phosphorene is ultraflat like graphene. If this is the case, they should observe by STM the first nearest P atoms. In all their images, only the second nearest P atoms were observed. This is a signature of buckling.*

In addition, they gave a unit cell distance of 4.41 Å, which gives a P-P first nearest atoms distance of 2.54 Å. This value is very big compared to the expected one (2.25 Å).

This is probably because the authors used Cu(111)-($\sqrt{3} \times \sqrt{3}$)R30° superstructure instead of Cu(111)-0.8($\sqrt{3} \times \sqrt{3}$)R30° reported in ref 20.

Responses: The high-resolution STM image of Fig. 1g in the original manuscript is indeed not clear enough to illustrate the atomic structure of the ultraflat BlueP. Here, we have performed additional experiments to obtain the atomic-resolution STM image. Figure R8b is the new result of the atomic-resolution STM image. As marked by the blue balls, the first and second nearest P atoms are clearly observed, demonstrating that the chiral BlueP holds an ultraflat honeycomb structure like graphene rather than a buckled structure like stanene. Here, for more clear illustration of the ultraflat honeycomb structure, we have replaced the atomic-resolution STM image in Fig. 1g (see Fig. R9).

Separately, our theoretical calculation shows that when the unit cell distance of BlueP increases up to 3.91 Å, the buckled structure will totally transform into the ultraflat structure like graphene (see Fig. S30), consistent with another previous work [ACS Nano 14, 2385-2394 (2020)]. If we continue to increase the unit cell distance, the BlueP also keeps an ultraflat structure then the distance of the P-P first nearest atoms will increase (larger than 2.25 Å). Our RHEED and STM results show that the ultraflat chiral BlueP has a unit cell distance of ~4.18 Å (see Fig. 1c and 1g), larger than the value of buckled BlueP [~3.5 Å, Adv. Funct. Mater. 33, 2213664 (2023)]. This unit cell distance gives a P-P first nearest atoms distance of 2.41 Å (see Fig. R9b). Though it is larger than the predicted value (buckled BlueP, 2.25 Å), it is consistent with the calculations (see Fig. S32), as when the unit cell distance is larger than 3.91 Å, the P-P nearest neighboring distance will increase (larger than 2.25 Å).

Figure. R8 a, High-resolution STM image of ultraflat chiral BlueP. **b**, Atomic resolution STM image corresponding to **(a)**, showing an ultraflat honeycomb structure like graphene. The blue balls mark the position of phosphorus atoms and the white diamond indicates the unit cell of ultraflat chiral BlueP.

Figure. R9 High-resolution STM image of ultraflat chiral BlueP. The black diamond marks the unit cell of superstructure. The inset is the zoom-in atomic-resolution STM image ($2 \times 2 \text{ nm}^2$), clearly showing an ultraflat honeycomb structure like graphene. The blue balls mark the phosphorous atoms and the yellow diamond labels the unit cell of ultraflat BlueP.

Comment 4: *I don't understand why the authors performed ex-situ XPS measurements on phosphorene. It is well known that phosphorene is not stable in air. This doesn't add any added value to the manuscript. In-situ measurements are necessary to explain the different structures.*

Responses: In retrospect, this part of the ex-situ XPS measurements was done in response to a suggestion raised by Reviewer 3 in a separate review. In the past, the stability of the BlueP on Au(111) has been investigated. The ultraflat chiral BlueP studied here is a new phosphorene structure that is different from the phosphorene grown on Au(111) and may show different stability. Therefore, we also performed the XPS measurements to investigate its stability. Figure S2 shows the ex-situ characterization result of XPS for the ultraflat chiral BlueP sample. The XPS spectrum shows only the P-O peak but not the pure P peak, indicating that the ultraflat chiral BlueP is vulnerable to environmental degradation as well. Therefore, for some practical applications, proper encapsulation [as demonstrated by Li et al., Nat. Nanotechnol. 10, 227-31 (2015)] is needed. For in-situ measurements, it is a good suggestion to investigate the differences between the different phosphorene structures. However, limited by our instrument capacities, we have not performed such experiments. In the future, we will try to collaborate with other groups to perform this study.

In response to the Reviewer's comment, on page 9 in the main text, the sentence "**As the XPS result shows, it is vulnerable to environmental degradation (see Supplementary Fig. 2); therefore, for some practical applications, proper encapsulation will be needed [Ref. 47].**" has been added.

Detailed responses to Reviewer 3

Generic Comments: *The manuscript by Song et al. reports on the macroscale area epitaxy of ultra flat blue-phosphorene on (111) terminated Cu substrate with asymmetric bimodal structure displaying a chiral character. The emergence of the ultra-flat and chiral character as salient features of the so developed epitaxy are explained in terms of the subtle interaction of the P atom arrangement with the substrate. Based on scanning tunnelling spectroscopy, the so grown blue phosphorene exhibits a local metallic character. Structural variations in the phosphorene structure are also investigated as a function of the growth condition in order to disclose an exhaustive number of achievable motifs. Getting through the scalability of 2D phosphorus is still challenging and deserves consideration, but I have some major (1-3) and technical (4-9) concerns on the present submission as follows that should be addressed to have the manuscript reconsidered.*

Responses: We thank the Reviewer for the succinct summary of the main findings and the support of this work. In the following, we reply to the various comments and suggestions point-by-point.

Comments 1: *For the sake of transparency, I already happened to review the present manuscript for another journal with more stringent standards of publication. After that stage, I realise the present submission includes some little changes consisting of the inclusion of Figure 5 as an overview of the possible phosphorene structures, and of Ref. 20 by Kaddar et al. Ref. 20 firstly reported on the large scale epitaxy of blue-phosphorene on Cu(111) methodologically in complete analogy with the present work. This is a strong limitation to the prior novelty of the work if the Authors aim at a high impact publication. However, the progress on the phosphorene growth are demanding in consideration of the fact that black phosphorus is hardly scalable (hardly but not impossibly, as reported in Nature Materials 20, 1203 (2021)) and the present work is step beyond the current knowledge on the matter. However, while in Ref. 20 phosphorene is observed to be a Dirac semimetal, the present manuscript reports on the metallic character of the blue-phosphorene on Cu(111) as derived from local scanning tunnelling spectroscopy. To avoid a conflict of statements, this discrepancy shall be mandatorily discussed and elaborated in a revised version of the present submission. In the end, it should be clear in what terms the cases differentiate and what is the true electronic character(s) of the epitaxially grown phosphorene on Cu(111).*

Responses: For freestanding BlueP, the calculated band structure and density of states reveal that it is a semiconductor with an indirect band gap of about 2 eV [Ref. 9, Phys. Rev. Lett. 112,

176802 (2014)]. In addition, its band gap also depends sensitively on the applied in-plane strain. As the BlueP is compressed or stretched in-plane, its band gap will decrease. When the stretch is large enough, the band gap will close, resulting in the transformation into metallic BlueP. In our experiment, the ultraflat chiral BlueP has a lattice period of ~ 4.18 Å, stretched by 30% compared with 3.26 Å (expected value for freestanding BlueP). According to our calculations, such ultraflat BlueP already shows a metal band structure in vacuum (without substrate, see Fig. S33a), which is consistent with the previous work [ACS Nano 14, 2385-2394 (2020)]. When considering the Cu(111) substrate (see Fig. S33b), the system also keeps a metallic nature, where the hybridized P-Cu band crosses the Fermi energy. Our STS results indeed show a metallic electronic structure, consistent with the DFT results. For the buckled BlueP grown on Cu(111) in the recent work [Adv. Funct. Mater, 33, 2213664 (2023)], its lattice (period ~ 3.4 Å) is stretched only mildly compared with the freestanding BlueP, and it still maintains the buckled honeycomb structure. DFT calculations [Phys. Rev. Lett. 112, 176802 (2014)] revealed that such an isolated buckled BlueP (without substrate) is still a semiconductor with a large indirect band gap. In their work, however, a Dirac cone was observed by ARPES, and the STS acquired on the phosphorene layer indicates a metallic character as there is no evidence of a band gap, in contrast to what is usually expected for isolated buckled BlueP [Phys. Rev. Lett. 112, 176802 (2014)]. This indicates that the substrate has a fundamental impact on its intrinsic properties. Their supporting LDOS results also revealed that the buckled BlueP on Cu(111) exhibits a metallic character, which contrasts with its corresponding isolated buckled BlueP, or buckled BlueP on Au(111) [ACS Nano 12, 5059-5065 (2018)]. They attributed this peculiar behavior to a specific hybridization between the P and Cu electronic states.

In response to the Reviewer's comments, on page 6 in the main text, the sentences **“When we focus on a narrow energy-range dI/dV spectrum (see Fig. 4b), one can find that the DOS around Fermi energy is far from zero, suggesting a metallic nature, consistent with our calculations (see Supplementary Fig. S24) and the previous theory.”** have been expanded to **“The small-energy scale STS (see Fig. 4b) indicates a metallic character for ultraflat chiral BlueP. To further reveal its electronic structure, we have also performed DFT calculations. As shown in Supplementary Fig. S33a, the band structure of an isolated ultraflat BlueP shows a metallic character as its lattice has a large in-plane stretch, consistent with the previous study that the bandgap decreases with the increase of the strain [Ref. 9]. When considering the Cu(111) substrate (see Supplementary Fig. S33b), P and Cu hybridized bands appear near the Fermi level and it also keeps a metallic character. Also, a recent work [Ref. 21] reported the fabrication of buckled BlueP with a lattice period of ~ 3.4 Å on Cu(111). In that work, it was found that the isolated buckled BlueP holds a large band gap. However, their experimental and DFT results show that the buckled BlueP on Cu(111) displays a metallic character, which was attributed to specific hybridization between the P and Cu electronic states. In contrast, the BlueP on**

Au (111) exhibits a semiconductor bandgap [Ref. 35].”

Comments 2: *Telling the true electronic character is also critical to assess the impact of the blue-phosphorene itself (and even its name, does it make sense to name it blue-phosphorene if it is metallic? Perhaps chiral / epitaxial phosphorene is more appropriate) in the full framework of the Xenes. Indeed, the evidence of a purely metallic nature limits the significance of the materials itself with respect to the wealth of more intriguing and technology relevant configurations that can be found in the Xenes (see for instance the topological character in stanene, bismuthene, the semi metallic character in silicene, etc.). In origin, blue phosphorus, namely rhombohedral phosphorus, is expected to be semiconducting even at its 2D state (Phys. Rev. Lett. 112, 176802). It is then plausible to assign its metallic character to the interaction with its substrate as happens with other Xenes like silicene (see Phys Rev B 92, 165427 (2015)). It would be interesting to assess to what extent the metallic character is induced by proximity with the metal substrate and how this proximity can be alleviated or removed.*

Responses: Zhen Zhu et al. first predicted a new phosphorus allotrope with a buckled honeycomb structure [Phys. Rev. Lett. 112, 176802 (2014)] and named it “blue phosphorus” as its largest band gap corresponds to the photon energy of visible blue light. The band gap of blue phosphorus depends on layer numbers and in-plane stress. When its band gap does not correspond to the photon energy of visible blue light, they still named it “blue phosphorus”. Phosphorus has many allotropes, including black phosphorus, white phosphorus, red phosphorus, blue phosphorus, and so on. Each phosphorus allotrope holds its unique crystalline structure. For example, the black phosphorus is an orthorhombic structure made up of puckered layers, and the blue phosphorus shows a rhombohedral structure with buckled honeycomb layers. Therefore, the blue phosphorus represents the phosphorus allotrope with a honeycomb structure in layers. In our manuscript, we named the chiral phosphorene “chiral blue phosphorene” as it also holds a honeycomb structure but without considering its electronic properties. As reported in previous works [2D Mater. 1, 031002 (2014), Adv. Funct. Mater. 33, 2213664 (2023)], the authors also used the name “blue phosphorene” to describe the metallic honeycomb phosphorene structure. Therefore, we would prefer to keep the term “blue phosphorene” in the present manuscript to provide readers with a first-glance understanding of the honeycomb structure of chiral phosphorene.

For phosphorene on Au(111), as reported earlier, it also holds a semiconductor band gap of about 1 eV [ACS Nano 12, 5059-5065 (2018)] without emerging hybridized band with the substrate, indicating weak interaction with the substrate. Due to the stronger chemical activity of the Cu(111) surface compared with Au(111), the Cu(111) surface would significantly alter the intrinsic properties of the epitaxial thin film, such as the graphene/Cu(111). Both our work and that of Kaddar et al. [Adv. Funct. Mater. 33, 2213664 (2023)] reveal that the coupling between the ultraflat/buckled BlueP and Cu(111) substrate is non-negligible, resulting in the

hybridized bands with the substrate near the Fermi level. This is similar to the case of silicene on Ag(111) [Phys Rev B 92, 165427 (2015)], which exhibits a strong Si-Ag hybridization. Though the BlueP/Cu(111) shows metallic character, as demonstrated in the work of Ref. Adv. Funct. Mater. 33, 2213664 (2023), a large Fermi velocity is revealed for the phosphorene layer of $V_F = 0.86 \times 10^6 \text{ ms}^{-1}$ comparable to the one found for graphene. This large Fermi velocity implies a high carrier mobility. Thus, the metallic BlueP with high carrier mobility could be developed for transparent electrode.

In addition, to alleviate or remove the coupling from the metal substrate and make the properties of pristine BlueP accessible, a delicate tuning of the interfacial interaction is required. For instance, intercalation of hydrogen [Nature 577, 204-208 (2020)] or oxygen [ACS Nano, 6, 9951–9963 (2012)] has been successfully used for the electronic decoupling of graphene from the underlying substrate to restore its linear π -band dispersion. In the future, one can intercalate different atoms or molecules at the ultraflat chiral BlueP/Cu(111) interface to recover its intrinsic electronic properties.

In response to the Reviewer's comments, the sentence "**which could be developed for transparent electrode.**" has been added on page 6 in the main text. And the sentence "**This is similar to the silicene grown on Ag(111) [Ref. 34], which exhibits a strong Si-Ag hybridization.**" has been added. In addition, the description "**In order to alleviate or remove the coupling from metal substrate and make the properties of pristine BlueP accessible, in the future, one can intercalate different atoms or molecules at the BlueP/Cu(111) interface to recover its intrinsic electronic properties, as demonstrated in graphene/metal systems [Refs. 36 and 37].**" has been added.

Comments 3: *The evidence of a chiral structure in the so-grown phosphorene is intriguing but at the present there is no implication of chirality in any potentially exploitable physical property (for instance, optical dichroism, or so). This makes it an interesting result of the 2d growth with a short momentary impact. I recommend the Authors to go through the potential of the chiral structure in terms of possible implications or at least, discuss what they prospect to be gained by taking benefit from the chiral phosphorene.*

Responses: To further enhance the expected impact of this work, we have broadened the discussions on its significance. Our present discovery may have far-reaching fundamental and practical implications. At a fundamental level, we believe the reversible transformation between the chiral and achiral phases of ultraflat BlueP offers new insights into the ever-inspiring secrets of emergent chirality in nature. At a practical level, the demonstrated capability of controlled mass production of chiral templates may enable further developments of novel chiral functional platforms. As an example, chirality-induced spin selectivity in electron transmission has been reported experimentally in a thin chiral molecules layer grown

on metal substrates [Science 283, 814-816 (1999), Science 331, 894-897 (2011)], which could be used to generate highly polarized electron beams. As shown in Fig. S17 and S18, the STM morphology and surface electronic states exhibit a periodic spiral pattern related to the chiral superstructure, indicating that the surface electronic states of BlueP are strongly modulated by the chiral superstructure as well. This character implies that different chiral phases might lead to opposite spiral potentials [Adv. Mater 34, 2106629 (2022)] and show different responses to the incident polarized photons or spin-polarized electrons, which can be exploited for a range of potential applications in polarization optics and spintronics. In addition, as the surface plays an important role in many chemical processes such as adsorption, catalysis, and crystallization, developing chiral surfaces and understanding their enantioselectivity are potentially rewarding. Chiral surfaces possess enantioselective physical and chemical properties, making them suitable for many other enantioselective chemical and physical processes, including multiphase catalysis, enantioselective sensing, and crystallization [Nat Mater 19, 939-945 (2020), ACS Nano 4, 5-10 (2010)]. Specifically, chiral surfaces can provide an environment for various isomeric enantioselective processes used in industries such as pharmaceuticals and agricultural chemicals. The active nature of the chiral BlueP stabilized on Cu(111) might be essential in rendering the system highly desirable for enantioselective chemical processes. Therefore, the chiral BlueP could provide a new platform of chiral metal surface for studying chiral catalysis, chiral recognition, and separation.

In response to the Reviewer's comments, we have revised several parts in the main text as follows:

In Abstract, the sentence **“Introducing chirality into BlueP can further enrich its physical properties and expand the application potentials.”** has been revised to **“Introducing chirality into BlueP can further enrich its physical and chemical properties and expand the application potentials.”**. The sentence **“which can be exploited for a range of potential applications including chiral catalysis.”** has been revised to **“which can be exploited for a range of potential applications including polarization optics, spintronics, and chiral catalysis.”**.

On page 2 in the main text, the sentence **“Chiral materials that break the mirror symmetry and show different responses to externally incident left-handed or right-handed polarized light have promising applications in optoelectronics, spintronics, and valleytronics.”** has been revised to **“Chiral materials that break the mirror symmetry and show different responses to externally incident left-handed or right-handed polarized light have promising applications in optoelectronics, spintronics, valleytronics, and chiral catalysis.”**. The sentence **“Evidently, introducing chirality into BlueP can further enrich its physical properties and expand the application potentials.”** has been revised to **“Evidently, introducing chirality into BlueP can further enrich its physical and chemical**

properties and expand the application potentials.”.

On page 8 in the main text, we have been added the description “**At a practical level, the demonstrated capability of controlled mass production of chiral templates may enable further developments of novel functional platforms for applications in polarization optics, spintronics, and asymmetric catalysis. More specifically, we can envision developments of new types of chiral catalysts, exploiting the active nature of the BlueP monolayer in accommodating adsorbing molecules with pre-selected chirality.**” has been revised to “**The demonstrated capability of controlled mass production of chiral templates may enable further developments of novel chiral functional platforms. As an example, chirality-induced spin selectivity in electron transmission has been reported experimentally in a thin chiral molecules layer grown on metal substrates [Refs. 13 and 45], which could be used to generate highly polarized electron beams. As shown in supplementary Fig. 17 and Fig. 18, the STM morphology and surface electronic states exhibit the periodic spiral patterns related to the chiral superstructure, indicating that the surface electronic states of the BlueP are strongly modulated by the chiral superstructure as well. This character implies that different chiral phases might lead to opposite spiral potentials [Ref. 46] and show different responses to the incident polarized photons or spin-polarized electrons, which can be exploited for a range of potential applications in polarization optics and spintronics. In addition, as the surface plays an important role in many chemical processes such as adsorption, catalysis, and crystallization, developing chiral surfaces and understanding their enantioselectivity are potentially rewarding. Chiral surfaces possess enantioselective physical and chemical properties, making them suitable for many other enantioselective chemical and physical processes, including multiphase catalysis, enantioselective sensing, and crystallization [Refs. 15 and 48]. Specifically, chiral surfaces can provide an environment for various isomeric enantioselective processes used in industries such as pharmaceuticals and agricultural chemicals. The active nature of the chiral BlueP stabilized on Cu(111) might be essential in rendering the system highly desirable for enantioselective chemical processes. Therefore, the newly discovered chiral BlueP could provide a new platform of chiral metal surface for studying chiral catalysis, chiral recognition, and chiral separation.**”.

The comments 4-8 are the same as in the previous review on our submission to another journal, and have largely been addressed in the submitted manuscript to Nature Communications. In this further revised version, we have made additional corrections to improve the paper based on these comments. In the following, we choose to still reply to the comments and suggestions point-by-point.

Comments 4: *Fig. 1 is not satisfactory in giving an atomically resolved structure of the blue-phosphorene cell to be more closely matched with that debated in the case of the epitaxy on*

Au(111) where different models - Au-P framework [J. Phys. Chem. C 2020, 124, 2024] and adatom-rich structure [Nano Lett. 2016, 16, 4903] - were proposed to rationalise the local STM topography. Data on the atomic structure should more extensively shown;

Responses: Figure 1g in the original manuscript is indeed not clear enough to illustrate the atomic structure of the ultraflat BlueP. We have performed supplementary STM experiments to obtain a clearer atomic-resolution STM image of BlueP. As shown in Fig. R8b, the atomic-resolution STM image clearly shows that the chiral BlueP possesses an ultraflat honeycomb structure similar to graphene, rather than the other structures reported in the literatures, such as the Au-P framework [Zhao et al., J. Phys. Chem. C, 124, 2024 (2020)] or the adatom-rich structure [Zhang et al., Nano Lett. 16, 4903 (2016)]. Here, we have replaced the atomic-resolution STM image in Fig. 1g (see Fig. R9) to clearly illustrate the ultraflat honeycomb structure.

Figure. R8 **a**, High-resolution STM image of ultraflat chiral BlueP. **b**, Atomic resolution STM image corresponding to **(a)**, showing an ultraflat honeycomb structure like graphene. The blue balls mark the position of phosphorus atoms and the white diamond indicates the unit cell of ultraflat chiral BlueP.

Figure. R9 High-resolution STM image of ultraflat chiral BlueP. The black diamond marks the unit cell of the superstructure. The inset is the zoom-in atomic-resolution STM image ($2 \times 2 \text{ nm}^2$), clearly showing an ultraflat honeycomb structure like graphene. The blue balls mark the phosphorous atoms and the yellow diamond labels the unit cell of ultraflat BlueP.

Comments 5: *what is the physical meaning of the bright protrusions in Fig. 1g at the vertex of the spiral triangles? Even in this case, an atomistic picture of the phosphorene structure is due for the sake of a more exhaustive understanding.*

Responses: In Fig. 1g, the bright protrusions are located at the positions of the nodes. The atomic-resolution STM image (see Fig. R6) shows that such bright protrusions are composed of three P atoms (a phosphorous trimer). From the line-scan profile across the protrusions (see Fig. S4), the measured height of the protrusion is $\sim 0.8 \text{ \AA}$, indicating that the phosphorous trimers are located on top of the phosphorene layer. We note that most of the bright protrusions are phosphorous trimers, and only a small portion of them are dimers and tetramers (see Fig. 2). As proposed in the schematic diagrams of Figs. 3a and 3b, at the node, the first-layer atom is located at the on-top site, which is not a stable site. Therefore, the atoms in the nodal area relax via atomic displacement to minimize the total energy. As shown in Fig. R6c, there exists obvious misfit dislocations in the nodal area due to the atomic displacement, as guided by the black lines. Therefore, this chiral pattern is very different from the moiré spots reported in buckled BlueP grown on the Cu(111) substrate [Adv. Funct. Mater. 33, 2213664 (2023)].

In response to the Reviewer's comment, Figure S13 has been replaced by Fig. R6 in the Supplementary Information.

On page 4 in the main text, the sentence **“The atomic resolution STM image (see Supplementary Fig. 13) and line scan profile (see Supplementary Fig. 4) show that these bright protrusions at the nodes are composed of three P atoms (a P trimer) located on the phosphorene layer.”** has been added.

On page 3 in the Supplementary Information, we have added the sentences **“We note that there are bright protrusions at the nodes. The atomic resolution STM image (see Fig. S13) shows that these bright protrusions are composed of three P atoms (a P trimer). From the line scan profile across the protrusions (see Fig. S4), the measured height of the protrusion is $\sim 0.8 \text{ \AA}$, indicating that the phosphorous trimers are located on top of the phosphorene layer. Most of the bright protrusions are phosphorous trimers, and only a small portion of them are dimers and tetramers (see Fig. 2). As proposed in the schematic diagrams of Figs. 3a and 3b, at the node, the first-layer atom is located at the on-top site, which is not a stable site. Therefore, the atoms in the nodal area relax via atomic displacement to minimize the total energy. As shown in Fig. S13c, there exist obvious misfit dislocations in the nodal area due to the atomic displacement, as guided by the black lines. Therefore, this chiral pattern is very different from the moiré spots reported in buckled BlueP grown on the Cu(111).”**

Figure. R6 (reproduced) a,b, Atomic resolution STM images ($10 \times 10 \text{ nm}^2$, 78K) of ultraflat chiral BlueP. The green dots in **(b)** mark the P atoms at the nodes. **b**, Bragg-peaks filtered image corresponding to **(a)**, which can enhance the clarity of the lattice. The green dots mark the P atoms at the nodes, and the yellow dots mark the positions of ultraflat chiral BlueP lattices. There exists obvious misfit dislocation as guided by the black lines.

Comments 6: *no insight is given through the environmental stability of the so-grown phosphorene while it is known that buckled blue-phosphorene as well as puckered black phosphorene are vulnerable to environmental degradation [Nanoscale 2019,11, 18232 and Nano Lett. 2014, 14, 6964, respectively] and its encapsulation is compulsory for every deployment in application.*

Responses: Figure S2 shows the ex-situ characterization result of XPS for the ultraflat chiral BlueP. The XPS spectrum shows only the P-O peak but not the pure P peak, indicating that the ultraflat chiral BlueP is vulnerable to environmental degradation. Therefore, for some practical applications, proper encapsulation [as demonstrated by Li et al., Nat. Nanotechnol. 10, 227-31 (2015)] is needed.

In response to the Reviewer's comment, on page 9 in the main text, we have added the sentence: **“As the XPS result (see Supplementary Fig. 2) shows, it is vulnerable to environmental degradation; therefore, for some practical applications, proper encapsulation is needed [Ref.47].”**

Comments 7: *while the origin of the chiral structure is attributed to the differential match with the substrate, it is not clear how the chirality of phosphorene motifs and hence, its twisted orientation with the substrate can be effectively controlled by a proper surface engineering or so. If this aspect is not clarified, since the epitaxy is not a controllable top-down scheme, I'm afraid that chiral structures as observed could turn out to be randomly emerging with no possible control over it thus limiting its applicability.*

Responses: We thank the Reviewer for the constructive comment. This comment is addressed together with comment 8 below.

Comments 8: *to what extent the chiral motifs in blue-phosphorene are statistically supported? And to what extent is the epitaxy of epitaxial phosphorene affected by growth parameter variability? For instance, what is the influence of the temperature in the formation of ultra-flat phosphorene and hence, in the emergence of the observed chiral motifs? Aiming at the dissemination through Science should entail a statistically extensive examination of the case in all the configurations to get rid of possible singular or local events. In the present case, the manuscript appears to be quite short in the extensive consideration of the experimental case which is reduced to a single, though interesting, configuration.*

Joint Responses to Comments 7 & 8: We fully agree and deeply appreciate these challenging questions, which have helped to largely improve characterization and understanding of the growth of the 2D chiral crystals. As the focused phosphorene structure in the manuscript, the ultraflat chiral BlueP cannot be directly obtained on Cu(111), but is obtained from the transition of other phosphorene structure as the coverage increase. In our experiment, we observed four different phosphorene structural phases as the phosphorous coverage increases, including the chain, strip, chiral, and hexagonal phases, as shown in Fig. 5. We found that the phosphorene structural phases grown on Cu(111) depends critically on the phosphorous coverage (here, 1ML phosphorene is defined when the surface is fully covered by ultraflat chiral BlueP) at a proper substrate temperature (200-250°C). First, when the coverage is below about ~0.85 ML, a chain phosphorene structure is formed. Figure R10a is the STM topographic image with a coverage of ~0.5 ML, showing a chain structure and bare Cu(111) surface coexist. When the coverage is up to ~0.85 ML, the Cu(111) surface is fully covered by the chain phase (see Fig. R10b). With more phosphorus deposited (> 0.85 ML), the chain phase will transform into a strip phase, as shown in Fig. R10c, where the chain and strip phases coexist. As the coverage further increases to ~0.95 ML, the chain phase will totally transform into the strip phase (see Fig. R10d). If we continue to increase the coverage (> 0.95 ML), the strip phase starts to transform into the chiral phase. Figure R10e shows the STM image after the deposition of ~0.97 ML phosphorus, where the strip and chiral phases coexist. When the coverage is up to 1 ML, the chiral phase would fully cover the Cu(111) surface (see Fig. R10f). As we continue to deposit phosphorous (>1 ML), the chiral phase slowly transits into the hexagonal phase (see Fig. R10g). In the end, the later-arrived P atoms are adsorbed on the nodes, forming self-assembled P nanodots (see Fig. R10h) located on the top of the phosphorene layer. As shown in Fig. R11, the apparent height of the P nanodots is ~2 Å, indicating the P nanodots are located on the top of the phosphorene layer. The measured period of the P nanodots is ~ 3.1 nm. The observed apparent height and the period of the P nanodots are consistent with the results of Ref. 20, Adv. Funct. Mater. 33, 2213664 (2023), suggesting that the self-assembled P nanodots are of the same structure on the ultraflat (here) and buckled BlueP.

Interestingly, we can also controllably decrease the P coverages through annealing. Under

proper annealing temperature ($\sim 350^\circ\text{C}$), the P atoms can slowly desorb from the phosphorene sheet, resulting in a decrease of coverage, and the direction of the phase transitions can be reversed as well (see Fig. 5). These experimental results indicate that the formation of all the phosphorene structures on Cu(111) sensitively depends on the coverage at a proper substrate temperature ($200\text{-}250^\circ\text{C}$). Through precise control of the coverage, one can select a single phosphorene phase on the Cu(111) surface. As shown in Fig. S19, the large-area ultraflat chiral BlueP fully covers the Cu(111) substrate surface, which is only limited by the size of our actual probe. When the substrate temperature is below 200°C , such as 180°C , multiple phases will coexist on the surface (see Fig. R12), including the chain, strip, and chiral phases. We suggest that the low surface temperature can cause a short migration distance for the P atoms or dimers to migrate on the surface, thus the different coverage in various areas is formed on the surface. Therefore, several phosphorene structures appear on the surface.

In response to the Reviewer's comments, on page 3 in the main text, we have been added the sentence **"In addition, we have achieved a reversible transformation between the chiral and achiral phases of ultraflat BlueP, which reveals a distinctive growth mode for phosphorene structures."** On page 7 in the main text, the sentence **"More details about the controllability of these competing phosphorene structures can be seen in Supplementary Information Note 6."** has been added.

In addition, we have added a new section to Supplementary Information as follows:

"Supplementary Note 6. The evolution of the phosphorene structures as the coverage varies.

As the focused phosphorene structure in the experiment, the ultraflat chiral BlueP cannot be directly obtained on Cu(111) but is obtained from the transition of other phosphorene structures as the P coverage increases. In our experiment, we observed four different phosphorene structural phases, including the chain, strip, chiral, and hexagonal phases, as shown in Fig. 5. We found that the phosphorene structural phases grown on Cu(111) depends critically on the phosphorous coverage (here, 1 ML phosphorene is defined when the surface is fully covered by ultraflat chiral BlueP) at a proper substrate temperature ($200\text{-}250^\circ\text{C}$). First, when the coverage is below about ~ 0.85 ML, a chain phosphorene structure is obtained. Figure S23a shows the STM topographic image with a coverage of ~ 0.5 ML, showing a chain structure and Cu(111) surface coexist. When the coverage is up to ~ 0.85 ML, the Cu(111) surface is fully covered by the chain phase (see Fig. S23b). Further depositing phosphorus (> 0.85 ML), the chain structure phase will transform into a strip phase, as shown in Fig. S23c, where the chain and strip phases coexist. As the coverage further increases to ~ 0.95 ML, the chain phase will totally transform into the strip phase (see Fig. S23d). If we continue to increase the coverage (> 0.95 ML), the strip

phase starts to transform into a chiral phase. Figure S23e shows the STM image after the deposition of ~ 0.97 ML phosphorus where the strip and chiral phase coexist. When the coverage is up to 1 ML, the chiral phase would fully cover the Cu(111) surface (see Fig. S23f). As we continue depositing phosphorous (>1 ML), the chiral phase slowly converts to the hexagonal phase (see Fig. S23g). In the end, the later-arrived P atoms adsorb on the node position, forming self-assembled P nanodots (see Fig. S23h) located on top of the phosphorene layer. As shown in Fig. S24, the apparent height of the P nanodots is ~ 2 Å, indicating that the P nanodots are located on the top of the phosphorene layer. The measured period of the P nanodots is ~ 3.1 nm. The observed apparent height and period of the P nanodots are consistent with the work of Kaddar et al. [Adv. Funct. Mater. 33, 2213664 (2023)], suggesting that the self-assembled P nanodots are of the same structure on the ultraflat (here) and buckled BlueP.

Interestingly, we can also controllably decrease the P coverage through annealing. Under proper annealing temperature ($\sim 350^\circ\text{C}$), the P atoms can slowly desorb from the phosphorene sheet, resulting in reversed phase transitions (see Fig. 5). These experimental results indicate that the formation of all the phosphorene structures on Cu(111) sensitively depends on the coverage at a proper substrate temperature (200 - 250°C). Through precise control of the coverage, one can select a single phosphorene phase on the Cu(111) surface. As shown in Fig. S19, the large-area ultraflat chiral BlueP fully covers the Cu(111) substrate surface, which is only limited by the size of our actual STM probe. When the substrate temperature is below 200°C , such as 180°C , multiple phases will coexist on the surface (see Fig. S25), including the chain, strip, and chiral phases. We suggest that the low surface temperature can cause a short distance for the P atoms or dimers to migrate on the surface, resulting in different coverage of various areas on the surface. Therefore, several phosphorene structures appear on the surface.”

Figure. R10 STM topographic images after growth of ~ 0.5 ML, ~ 0.85 ML, ~ 0.9 ML, ~ 0.95 ML, ~ 0.97 ML, ~ 1.0 ML, ~ 1.1 ML, and ~ 1.7 ML phosphorene on Cu(111).

Figure. R11 a, STM image of the phosphorene sheet at the coverage of 1.7 ML, highlighting the formation of the self-assembled P nanodots on its top. **b**, Line-profile along the blue line as indicated in the image, showing that the apparent height of the self-assembled P nanodots is ~ 2 Å, and the period of the P nanodots is ~ 3.1 nm.

Figure. R12 STM topographic image after depositing 0.8 ML phosphorous at ~ 180 °C, showing the coexistence of multiphases of phosphorene, including the chain, strip, and chiral phases.

Comments 9: *While commenting on Fig. 5 the Authors state that “various comprehensive changes in the interfacial interaction ultimately dictate the selection of the new structural phases”. It should be expected to understand how the variable interaction with the substrate affects the electronic character of the phosphorene. I recommend the Authors to elaborate more on this aspect.*

Responses: As discussed above, the interactions between the P-P atoms and between P-Cu atoms can be different for different phosphorene phases. From the chain phase to the hexagonal phase, the surface density of phosphorus atoms increases (see Fig. 5e-h), the bond length of P-P atoms becomes shorter, and the interaction of P-P atoms becomes stronger. At the same time, the binding between P and Cu atoms becomes weaker. We can also determine the P-Cu interaction strength from the stability of different phases through annealing experiments without the deposition flux, as shown in Figure. R13. We found that the hexagonal phase is the most unstable, with phosphorus atoms desorbing above ~ 250 °C. For the chiral phase, it stays stable at ~ 300 °C without phosphorus atoms desorbing or phase transition. The stripe phase can remain stable at ~ 320 °C, and the chain phase is the most stable structure, which can remain stable without phosphorus atoms desorbing even upon annealing it at 350 °C for a long time. The stability results of these different phosphorene structures are consistent with our suggestion that the most stable phosphorene structure corresponds to the strongest binding with the substrate. The influence of the Cu(111) substrate on the electronic structure of phosphorene should be closely related to the strength of the P-Cu interaction. Therefore, we suggest that the chain phase has the strongest hybridization between the P and Cu electronic states, while the hexagonal phase has the weakest hybridization. The STS results (see Fig. R14) show that all these phosphorene phases are of a metallic character. It is difficult to quantitatively determine their specific interaction with the substrate from the STS results. To further understand how the

electronic structures of different phases are affected by the substrate, further ARPES experiments and DFT calculations are needed.

In response to the Reviewer's comments, Fig. R13 and R14 have been added to the Supplementary Information as new Figs. S26 and S27. In addition, on page 7 in the main text, we have added the following sentences:

“The interaction strength of P-Cu atoms for different phosphorene phases is also revealed by the annealing experiments (see Supplementary Fig. 26), which shows that the chain phase is the most stable and the hexagonal phase is the most unstable, consistent with our analysis based on the bond length. Qualitatively, the stronger interaction with the substrate should induce a stronger effect on the electronic states of phosphorene by the substrate. Therefore, we suggest that the chain phase has the strongest hybridization between the P and Cu electronic states, and the hexagonal phase has the weakest hybridization. The STS results (see Supplementary Fig. 27) show that all these phosphorene phases are of a metallic character, making it difficult to quantitatively determine their specific interaction with the substrate from the STS results. To further understand how the electronic structures of different phases are affected by the substrate, further ARPES experiments and DFT calculations are needed.”

Figure. R13 Stability of different phosphorene structures under annealing. **a-d** STM images of the hexagonal, chiral, strip, and chain phases annealed at 250, 300, 320, and 350 °C, respectively, without the desorption of phosphorus atoms for each structure, signifying their stability.

Figure. R14 Large-energy scale STS taken on different phosphorene structures, including the chain, strip, chiral, and hexagonal phases, showing a metallic character for all the phases.

Finally, we wish to sincerely thank all the three reviewers again for their careful and expertized reviews, as well as for their constructive comments and suggestions. We hope this revised manuscript with much enhanced validity and clarity is now acceptable for publication in Nature Communications.

REVIEWER COMMENTS

Reviewer #1 (Remarks to the Author):

The authors have properly replied all my comments, and made significant improvement. The authors also provided further experimental results to support their findings and enhance the novelty of their work. Following are several minor issues based on the revised manuscript.

1. Seeing from the STM images, it shows an etching growth mode of P on Cu(111). Therefore, there is one possibility that the observed structure is a P-Cu alloy.
2. What are the atomic structures of the intermediates and other linear phases, e.g. Figure S23 and S26? Any DFT results?
3. In Figure S34, why the P pairs show a buckled structure?
4. In Figure S25, the coverage is 0.8 ML. However, there are many islands on the surface, which should appear at ~ 1.7 ML.
5. In Figure S23, what is the substrate temperature? The hexagonal phase in panel g and h is distinctly different. The spots in panel g should be the same as Figure 5d, while those bright dots are P clusters at high coverage.
6. In the caption of Figure S19, the authors mentioned a Moire pattern in the R area. According to the low-resolution STM image of panel c, the Moire pattern is hardly visible. Moreover, the direction of arrows in L and R (panel d and e) shows a relative angle of 13° . However, from the large scale STM image in panel a, this angle is $\sim 25^\circ$.

Reviewer #2 (Remarks to the Author):

The authors have modified the manuscript and added new data. They also answered all my questions/comments.

Reviewer #3 (Remarks to the Author):

In the resubmission the Authors closely addresses all the remarks that I have risen with satisfactory arguments. I think the novelty of the work is well-supported in terms of phase variability of epitaxial blue phosphorene on Cu(111) as a function of the kinetically limited parameters. The resubmitted manuscript has been substantially revised (mostly in the Suppl. Info. while minor changes are on the main body of the manuscript) and new arguments/clarifications have been specified so as to provide a more clear phenomenological picture. Adjustment of the title is also well-received and sound with the bulk of the

work. I still have some remarks on a highly improved manuscript as follows.

1- (mandatory) According to my Comment 1, it is clear (even from the other review) that the match with the paper by Kadar et al in Ref. 21 is critical to understand the novelty and the significance of the present work. It is also clear that the phase revealed by Kadar et al is not reproduced in the present work although the process conditions are quite similar despite possible variations from lab to lab. One key-ingredient that distinguished the works is the nature of the evaporants, P2 in the present case (as resulting from InP evaporation) and P4 in the paper by Kadar et al (resulting from bP dissociation). I think the Authors should more elaborate on the physical reason why a different “growth precursor” in an epitaxial growth may trigger different phase nucleation so as to give a more reconciling picture of the whole growth physics and thus explain why an apparently similar process results in a different structural scenario. I'm afraid this aspect should be mandatorily addressed even by spotting out sharp differences with the previously published paper by Radar et al.

2- (optional) According to my Comment 3 and 6 (also in relation to Fig. S2), mentioned applications about chirality sensitive spintronics and asymmetric catalysis are indeed interesting as perspectives but they should be grounded on a compliant stabilisation scheme without which the highly unstable blue-phosphorene turns out to be unpractical. In addition to the work by Li et al. Nature Nanotechnol. 10, 227 (2015) as reported in the Response, let me suggest the Authors to optionally consider material selective scheme for encapsulation like those implemented in case of silicene by Molle et al, Faraday Discuss. 227, 171 (2021), black Phosphorus by Wood et al, Nano Lett. 14, 6964 (2014), or blue-phosphorene on Au by Grazianetti et al, Nanoscale 11, 18232 (2019) as they may offer viable solutions of material exploitation out of the vacuum condition of the growth environment.

On top of these remarks, I think the manuscript extensively reports on a new fashion of temperature controllable phosphorene phases as never done before. It is not clear how to make use of these phosphorene allotropes in practical applications, nonetheless here one can find the details on how to produce a chiral phosphorene in a feasible processing scheme for further progress on the emerging chirality-sensitive materials applications. Therefore, I am pleased to consider the paper as suitable for dissemination through Nature Communications after the Authors will be able to address my present Comment 1 more specifically.

Responses to Reviewers' Second Reports
(MS # NCOMMS-23-33683B by Yeheng Song et al.)

The detailed responses are listed below, and the manuscript has been revised accordingly. With these responses and revisions, we hope that this paper is now ready for publication in Nature Communications.

Detailed responses to Reviewer 1

Generic Comments: *The authors have properly replied all my comments, and made significant improvement. The authors also provided further experimental results to support their findings and enhance the novelty of their work. Following are several minor issues based on the revised manuscript.*

Responses: We thank the reviewer for supporting this revised manuscript. In the following, we reply to the remaining comments and suggestions point-by-point.

Comment 1: *Seeing from the STM images, it shows an etching growth mode of P on Cu(111). Therefore, there is one possibility that the observed structure is a P-Cu alloy.*

Responses: At earlier stages of this study, we also suspected that the phosphorene structure on Cu(111) could be a P-Cu alloy as the growth behavior looks like an etching growth mode (see Fig. 5a), in which the phosphorene islands exhibit a lower step height than the Cu(111) terrace. But later on, we found that the step height of 2D phosphorene island depends on the bias voltage in our experiments. In STM, the tunneling current depends strongly on the local density of states (LDOS) of a surface; therefore, the measured height of a surface step is valid only when the upper terrace and the lower terrace have the same LDOS, i.e., they are the same material. In the case of phosphorene islands on the Cu(111) surface, the LDOS is quite different for the phosphorene sheet and the Cu(111) surface. As a result, the measured step height of the phosphorene island drastically depends on the STM bias voltage. To elaborate on this aspect, we show a series of STM images of the same area with different bias voltages (see Fig. R1). With a bias below 2.0 V, the phosphorene islands appear lower than the Cu(111) terrace

(see Fig. R1c-e). While the phosphorene islands are higher than the surrounding Cu(111) terrace (see Fig. R1a-b) with a bias voltage larger than 2.0 V, which does not like an etching growth mode. This phenomenon has been observed in previous work on Borophene [Nat. Chem 8, 563-568 (2016)], which was attributed to the low surface LDOS of Borophene compared to the metallic substrate terrace. In addition, from the high-resolution STM image of chiral BlueP (see Fig. 1d), an ultraflat honeycomb structure similar to graphene is clearly visible, in which all the atoms show the same characteristic, indicating that they belong to same type of atoms (P atoms). Based on these observations and the similarity of these phosphorene structures (see also the responses to Comment 2), we conclude that the observed phosphorene structure on Cu(111) is not a P-Cu alloy.

In response to the Reviewer's comment, we have added Fig. R1 to the Supplementary Information as new Fig. S39.

Figure. R1 a-e, A series of STM images of the same area with different bias voltages for the chain phase of phosphorene on Cu(111). **f**, The extracted step height of phosphorene islands from different tip bias STM images. The phosphorene islands are higher than the surrounding Cu(111) terrace (**a-b**) with a bias voltage larger than 2.0 V. While below 2.0 V, the phosphorene islands appear lower than the Cu(111) terrace (**c-e**). The bias-dependent height demonstrates the strong influence of the local density of states on the height measurement by STM.

Comment 2: *What are the atomic structures of the intermediates and other linear phases, e.g. Figure S23 and S26? Any DFT results?*

Responses: In our experiments, four phosphorene structures are observed, including chain, strip, chiral, and hexagonal phases (see Figs. S23 and S26). In the manuscript, an ultraflat honeycomb structure for the chiral phase is principally demonstrated by the high-resolution STM image and DFT calculations (see Fig. 1). According to the high-resolution STM images of phosphorene structures (see Fig. 5e-h), we marked the unit cells of different phosphorene structures. One can see that the chiral and hexagonal phases hold a similar hexagonal unit cell. The strip phase has a pseudo-hexagonal unit cell (oblique unit cell) that is very close to the chiral phase. The unit cell of the chain phase shows a large difference from the other three phases at first glance. However, when carefully comparing it with the strip phase, one can find that it is close to the 2×1 unit cell of the strip phase. Based on these results, we proposed the atomic structures of the other three phosphorene phases. As shown in Fig. R2, all the phosphorene structures show ultraflat honeycomb or pseudo-honeycomb structures. Specifically, the chiral and hexagonal phases have an ultraflat honeycomb structure like graphene; the strip phase shows a pseudo-honeycomb structure that is close to the chiral phase; and the chain phase can be regarded as a further deformation of the pseudo-honeycomb structure of the strip phase. Based on these proposed atomic structures, one can obtain the surface atomic density for different phosphorene phases and then obtain the coverages of the different phosphorene phases that fully cover the Cu(111) surface. Table R1 displays the coverages of different phosphorene phases estimated from the proposed atomic structures and obtained from the experiments. One can see that the calculated coverages are consistent with the experimental results, further supporting our proposed models. Since these phosphorene phases have similar atomic structures (see Fig. R2), they can easily undergo structural evolution as the coverage increases, or reverse as the coverage decreases (see Fig. 5).

The proposed atomic structure of the chiral phase is well consistent with the high-resolution STM image (see Fig. 1g), which all show an apparent ultraflat honeycomb structure akin to graphene. For the other three phosphorene phases (including chain, strip, and hexagonal phases), the high-resolution STM images cannot distinguish all the P atoms but only show bright protrusions in hollow sites (see Fig. 5e, f, h). This phenomenon has been reported in previous works of Borophene on Ag(111) [Nat. Chem. 8, 563-568 (2016) and Nat. Mater. 21,

35–40 (2021)], in which the high-resolution STM images cannot distinguish the B atoms due to the influence of the LDOS. As elaborated earlier, in STM, the tunneling current depends strongly on the LDOS of a surface, and therefore, the measured height of a local area may not represent the actual height but the LDOS strength. To fully determine the atomic structure of these phosphorene phases, atomic-resolution images are needed. The q-plus AFM can provide sub-atomic resolution images for surface structures, which are not influenced by the LDOS. In addition, it is also necessary to simulate these structures by DFT calculations, but such calculations are challenging to complete, especially given the reduced symmetries of the other three phases and correspondingly larger supercell sizes. It should be desirable to reveal the structural evolution of the phosphorene phase as revealed in detail by high-resolution q-plus AFM and corresponding DFT calculations in a future study.

In response to the Reviewer's comment, Fig. R2 and Table R1 have been added to the Supplementary Information as new Fig. S40 and Table S1. In addition, on page 8 in the Supplementary Information, the description about the proposed atomic structures of different phosphorene phases has been added as **“For the chiral phase, an ultraflat honeycomb structure similar to graphene is demonstrated by the high-resolution STM image and DFT calculations (see Fig. 1). According to the high-resolution STM images of phosphorene structures (see Fig. 5e-h), we mark the unit cells of different phosphorene structures. One can see that the chiral and hexagonal phases hold a similar hexagonal unit cell. The strip phase has a pseudo-hexagonal unit cell (oblique unit cell) that is very close to the chiral phase. The unit cell of the chain phase shows a large difference from the other three phases at first glance. However, as one carefully compares it with the strip phase, one can find that it is close to the 2×1 unit cell of the strip phase. Based on these results, we propose the atomic structures of the other three phosphorene phases. As shown in Fig. S40, all the phosphorene structures show ultraflat honeycomb or pseudo-honeycomb structures. Specifically, the chiral and hexagonal phases have an ultraflat honeycomb structure similar to graphene; the strip phase shows a pseudo-honeycomb structure that is close to the chiral phase; and the chain phase can be regarded as a further deformation of the pseudo-honeycomb structure of the strip phase. Based on these proposed atomic structures, one can obtain the surface atomic densities for different phosphorene phases and then obtain the coverages of the different phosphorene phases that fully cover the Cu(111) surface. Table S1 displays the coverage of different phosphorene phases estimated from the proposed atomic structures and obtained from the experiments. One can see that the coverages calculated by the proposed structures are consistent with the experimental results, further supporting our proposed models. Since these phosphorene phases have similar atomic structures (see Fig. S40), they can easily undergo structural evolution as the coverage increases, or reverse as the coverage decreases (see Fig. 5).**

The proposed atomic structure of the chiral phase is well consistent with the high-resolution STM image (see Fig. 1g), which shows an apparent ultraflat honeycomb structure akin to graphene. For the other three phosphorene phases (including chain, strip, and hexagonal phases), the high-resolution STM images cannot distinguish all the P atoms but only show bright protrusions in hollow sites (see Fig. 5e, f, h). This phenomenon has been reported in previous works of Borophene on Ag(111) [Nat. Chem. 8, 563-568 (2016) and Nat. Mater. 21, 35–40 (2021)], in which the high-resolution STM images cannot distinguish the B atoms due to the influence of the local electronic states (LDOS). In STM, the tunneling current depends strongly on the LDOS of a surface; therefore, the measured height of a local area may not represent the actual height but the LDOS strength. To fully determine the atomic structures of these phosphorene phases, atomic-resolution images are needed. The q-plus AFM can provide sub-atomic resolution images for surface structures without the influence from the LDOS. It should be desirable to reveal the structural evolution of the phosphorene phase as revealed in detail by high-resolution q-plus AFM and corresponding DFT calculations in a future study.”

Figure. R2 Proposed atomic structures for chain, strip, chiral, and hexagonal phases of phosphorene, respectively, showing an ultraflat pseudo-honeycomb in (a-b) or honeycomb structure (c-d).

Table. S1 Coverages of different phosphorene phases calculated by the proposed atomic structures and obtained from the experiments.

	Chain phase	Strip phase	Chiral phase	Hexagonal phase
Calculated coverage	0.91 ML	0.97 ML	1.0 ML	1.04 ML
Experimental coverage	0.85 ML	0.95 ML	1.0 ML	1.1 ML

Comment 3: In Figure S34, why the P pairs show a buckled structure?

Responses: We apologize for this confusion. In the original submission, the figure was slightly tilted, making it look like a buckled BlueP. To avoid potential future confusion, we have replaced Fig. S34 with Fig. R3.

Figure. R3 Cross-sectional electron localization function (ELF) showing high localization of the electrons in P–P pairs and weak P–Cu interaction.

Comment 4: In Figure S25, the coverage is 0.8 ML. However, there are many islands on the surface, which should appear at ~1.7 ML.

Responses: In Figure S25, the film is grown at 180°C. When the substrate temperature is below 200°C, such as 180 °C, multiple phases could coexist on the surface, including the chain, strip, chiral, and hexagonal phases. We suggest that, at the low surface temperature, the P atoms or dimers can migrate relatively short distances on the surface, resulting in different local coverages (local surface atomic densities of P atoms) of various areas. Therefore, several phosphorene structures could appear on the surface even when the coverage is below 1ML. That is, even though we only deposited 0.8 ML P on Cu(111), there could appear several phosphorene phases, in which some phases (such as chiral and hexagonal phases) can only appear at high enough coverages when the substrate temperature is larger than 200°C.

To better clarify the experimental results at low temperature, on page 8 in the Supplementary Information, we have modified as “**We suggest that, at the low surface temperature, the P atoms or dimers can migrate relatively short distances on the surface, resulting in different local coverages (local surface atomic densities of P atoms) of various areas. Therefore, several phosphorene structures could appear on the surface even when the coverage is below 1ML.**”

Comment 5: In Figure S23, what is the substrate temperature? The hexagonal phase in panel g and h is distinctly different. The spots in panel g should be the same as Figure 5d, while those bright dots are P clusters at high coverage.

Responses: In Figure S23, the substrate temperature is about 200 °C. From the large-scale STM images, Figs. S23g and S23h did show different characteristics. However, from the enlarged STM images (see Fig. R4), one can find that they all show a hexagonal superstructure (as a comparison, 1 ML phosphorene exhibits a chiral superstructure, as shown in Fig. 5c). Therefore, we named them all as the hexagonal phase in our manuscript. An evident difference between them is the structure located on the node. For 1.1 ML phosphorene, only several P atoms are located on the nodes, but for 1.7 ML phosphorene, there are P clusters located on the nodes (the same P nanodots structure as the Ref-20, Adv. Funct. Mater. 33, 2213664 (2023)). To avoid misleading readers, Fig. R4 has been added to the Supplementary Information as new Fig. S38. On page 7 in the Supplementary Information, we have the revised as “**As we continue to deposit phosphorous (>1 ML), the chiral phase slowly converts to the hexagonal phase (see Fig. S23g and Fig. S38a). In the end, the subsequently-arrived P atoms are adsorbed on the nodes, forming self-assembled P nanodots (as shown in Figs. S23h and Fig. S38b; we also name it hexagonal phase as it holds a hexagonal superstructure as well) located on the top of the phosphorene layer.**”. In addition, we have added the substrate temperature to the caption of Fig. S23.

Figure. R4 Enlarged STM images of 1.1 ML and 1.7 ML phosphorene on Cu(111). The yellow and green diamonds mark the unit cells of hexagonal superstructures. For 1.1 ML phosphorene, only several P atoms are located on the nodes, but for 1.7 ML phosphorene, there are P clusters located on the nodes (the same P nanodots structure as the Ref-20).

Comment 6: *In the caption of Figure S19, the authors mentioned a Moire pattern in the R area. According to the low-resolution STM image of panel c, the Moire pattern is hardly visible. Moreover, the direction of arrows in L and R (panel d and e) shows a relative angle of 13°. However, from the large scale STM image in panel a, this angle is ~25°.*

Responses: In taking pictures, when the spatial frequency of the pixels of the camera sensor is close to the spatial frequency of the stripes in the image, a new wavy interference pattern, the so-called Moiré pattern, may be generated. In our STM experiments, when the pixel resolution is close to the period of the chiral superstructure, a Moiré pattern appears in the STM image.

Therefore, the Moiré pattern observed in the large-scale STM image of Figure S19a is formed when we scan the STM image with low pixel resolution. If we scan with high pixel resolution, there will not appear a Moiré pattern (see Fig. S19d-e). Therefore, the Moiré pattern observed in the large-scale STM image does not correspond to the chiral superstructure. As shown in Fig. R5a, when the STM image is scanned with high pixel resolution, the chiral superstructure is clearly visible. However, it only shows a Moiré pattern formed by interference when the image holds a low pixel resolution (see Fig. R5b). Therefore, the direction of the Moiré pattern does not correspond to the direction of the chiral superstructure (as guided by the green and white arrows in Fig. R5).

Figure. R5 **a**, STM image in L area with 300×300 pixels. **b**, STM image with 30×30 pixels correspond to (a), showing a Moiré pattern formed by interference as guided by the green arrow, which does not correspond to the direction of the chiral superstructure as guided by the white arrow in (a).

Detailed responses to Reviewer 2

Generic Comments: *The authors have modified the manuscript and added new data. They also answered all my questions/comments.*

Responses: We thank the Reviewer again for assessing that the present manuscript is now acceptable for publication in Nature Communications.

Detailed responses to Reviewer 3

Generic Comments: *In the resubmission the Authors closely addresses all the remarks that I have risen with satisfactory arguments. I think the novelty of the work is well-supported in terms of phase variability of epitaxial blue phosphorene on Cu(111) as a function of the kinetically limited parameters. The resubmitted manuscript has been substantially revised*

(mostly in the Suppl. Info. while minor changes are on the main body of the manuscript) and new arguments/clarifications have been specified so as to provide a more clear phenomenological picture. Adjustment of the title is also well-received and sound with the bulk of the work. I still have some remarks on a highly improved manuscript as follows.

Responses: We thank the Reviewer for confirming the substantial improvement of the manuscript. In the following, we reply to the various comments and suggestions point-by-point.

Comments 1: *According to my Comment 1, it is clear (even from the other review) that the match with the paper by Kadar et al in Ref. 21 is critical to understand the novelty and the significance of the present work. It is also clear that the phase revealed by Kadar et al is not reproduced in the present work although the process conditions are quite similar despite possible variations from lab to lab. One key-ingredient that distinguished the works is the nature of the evaporants, P2 in the present case (as resulting from InP evaporation) and P4 in the paper by Kadar et al (resulting from bP dissociation). I think the Authors should more elaborate on the physical reason why a different “growth precursor” in an epitaxial growth may trigger different phase nucleation so as to give a more reconciling picture of the whole growth physics and thus explain why an apparently similar process results in a different structural scenario. I'm afraid this aspect should be mandatorily addressed even by spotting out sharp differences with the previously published paper by Radar et al.*

Responses: We appreciate the reviewer's pressing question for more mechanistic understanding of the different growth modes in the present study and Ref. 20. In response, we wish to dive into established understandings of growth phenomena for assistance, especially at the earlier stages of nucleation and growth. As we know, the initial stages of growth, characterized by the atomistic processes of adsorption, diffusion, nucleation, and islanding, play decisive roles in dictating the dominant growth mode(s) eventually selected [see, for example, Z. Y. Zhang and M. G. Lagally, Science 276, 377, 1997]. In this regard, we have explored the adsorption and diffusion properties of P adatom(s) on Cu(111), and the results are depicted in Figs. R6 and R7. Our detailed calculations reveal that a single P adatom prefers to occupy valley sites, as shown in Fig. R6a. When two P atoms are placed on the surface, they prefer to form a dimer with a P-P distance of 2.11 Å rather than staying apart (see Fig. R6). The calculated adsorption energies are -6.05 eV/P and -6.12 eV/P, respectively, for P monomer and dimer on Cu(111). The adsorption energy is defined as $E_{ads} = E_{total} - nE_P - E_{sub}$, where E_{total} , E_P , and E_{sub} are the total energies of the combined system, an isolated P atom in gas phase, and the Cu(111) substrate, respectively and n corresponds to number of P adatom(s). Furthermore, we have investigated the adsorption of isolated P₂ and P₄ on Cu(111), aiming to understand the role of the precursor in the nucleation and epitaxial growth of BlueP. The corresponding adsorption energy is defined as $E_{ads} = (E_{total} - E_{P_2/P_4} - E_{sub})$, where E_{total} and E_{sub} are the total energies of the combined system and pristine Cu(111) substrate, respectively, and

E_{P_2/P_4} corresponds to the total energy of an isolated P_2 or P_4 in the gas phase. The resulting E_{ads} are -1.67 eV/P and -0.99 eV/P, respectively, ensuring the stability of both P_2 and P_4 on the substrate.

Next, we have calculated the diffusion properties of both P_2 and P_4 in comparison to the P monomer on the Cu(111) substrate. In this regard, we explore various diffusion pathways for migration, as illustrated in Fig. R7. Our calculations reveal that the maximum diffusion barrier for a P monomer is 75 meV. A subtle and crucial aspect is that, considering the clear tendency of two P monomers to form a stable dimer, it is also possible that such a P - P dimer may diffuse faster as an entity than the separated monomers. Our detailed calculations indeed confirm this conjecture. As illustrated in Fig. R7, the minimum diffusion barrier for the P dimer is significantly reduced to 41.4 meV, even lower than that of P . The underlying reason of a stable and fast diffusing dimer on Cu(111) is inherently rooted in the relative binding strengths of P - P and P -substrate [Chen *et al.*, Phys. Rev. Lett. 104, 186101 (2010)]. Here, it is also worthwhile to mention that similar dimer feeding behaviors have been proposed for the initial growth stages of graphene on Cu substrates [P. Wu *et al.*, Phys. Rev. Lett. 114, 216102 (2015)], resulting in the growth of large-area and high-quality samples as observed [X. Li *et al.*, Science 324, 1312 (2009); X. Li *et al.*, Nano Lett. 9, 4268 (2009)]. More importantly, for a P_4 cluster, the diffusion barrier is much higher, at ~ 0.54 eV (Fig. R7). Therefore, we suspect that, in Ref. 20, the P_4 precursors have to be either first decomposed into P_2 dimers, then fast feed to the growing islands, or have to be supplied at a much slower rate as P_4 entities (see, for example, New. J. Phys. 10, 093026 (2008) & 11, 063046 (2009) for chain feeding in the case of graphene). Based on these calculations, we may conclude that two P atoms prefer to form a stable dimer on Cu(111) or P_2 dimers can be directly deposited on Cu(111), and such dimers are energetically stable and can diffuse much faster than either a P monomer or (especially) a P_4 cluster. Thus, the growth precursor (P_2 in the present case) in epitaxial growth may play a decisive role in the formation of ultraflat BlueP on Cu(111), as presented in the present study.

In response to the Reviewer's comment, we have added the following section about the atomistic mechanisms in the initial growth stages of ultraflat BlueP on Cu(111) to the Supplementary Information.

“Supplementary Note 8. The atomistic mechanisms in the initial growth stages of ultraflat BlueP on Cu(111).”

It is well established that the initial stages of growth, characterized by the atomistic processes of adsorption, diffusion, nucleation, and islanding, play decisive roles in dictating the dominant growth mode(s) eventually selected [Z. Y. Zhang and M. G. Lagally, Science 276, 377, 1997]. In this regard, we have explored the adsorption and diffusion properties of P adatom(s) on Cu(111) and the results are depicted in Figs. S36

and S37. Our detailed calculations reveal that a single P adatom prefers to occupy valley sites, as shown in Fig. S36a. When two P atoms are placed on the surface, they prefer to form a dimer with a P-P distance of 2.11 Å rather than staying apart (see Fig. S36). The calculated adsorption energies are -6.05 eV/P and -6.12 eV/P, respectively, for P monomer and dimer on Cu(111). The adsorption energy is defined as $E_{ads} = E_{total} - nE_P - E_{sub}$, where E_{total} , E_P , and E_{sub} are the total energies of the combined system, an isolated P atom in gas phase, and the Cu(111) substrate, respectively and n corresponds to number of P adatom(s). Furthermore, we have investigated the adsorption of isolated P₂ and P₄ on Cu(111), aiming to understand the role of the precursor in the nucleation and epitaxial growth of BlueP. The corresponding adsorption energy is defined as $E_{ads} = (E_{total} - E_{P_2/P_4} - E_{sub})$, where E_{total} and E_{sub} are the total energies of the combined system and pristine Cu(111) substrate, respectively, and E_{P_2/P_4} corresponds to the total energy of an isolated P₂ or P₄ in the gas phase. The resulting E_{ads} are -1.67 eV/P and -0.99 eV/P, respectively, ensuring the stability of both P₂ and P₄ on the substrate.

Next, we have calculated the diffusion properties of both P₂ and P₄ in comparison of P monomer on the Cu(111) substrate. In this regard, we explore various diffusion pathways for migration, as illustrated in Fig. S37. Our calculations reveal that the maximum diffusion barrier for a P monomer is 75 meV. A subtle and crucial aspect is that, considering the clear tendency of two P monomers to form a stable dimer, it is also possible that such a P-P dimer may diffuse faster as an entity than the separated monomers. Our detailed calculations indeed confirm this conjecture. As illustrated in Fig. S37, the minimum diffusion barrier for the P dimer is significantly reduced to 41.4 meV, even lower than that of P₄. The underlying reason of a stable and fast diffusing dimer on Cu(111) is inherently rooted in the relative binding strengths of P-P and P-substrate [Chen *et al* , Phys. Rev. Lett. 104, 186101 (2010)]. Here, it is also worthwhile to mention that similar dimer feeding behaviors have been proposed for the initial growth stages of graphene on Cu substrates [P. Wu *et al.*, Phys. Rev. Lett. 114, 216102 (2015)], resulting in the growth of large-area and high-quality samples as observed [X. Li *et al.*, Science 324, 1312 (2009); X. Li *et al.*, Nano Lett. 9, 4268 (2009)]. More importantly, for a P₄ cluster, the diffusion barrier is much higher, at ~0.54 eV (Fig. S37). Therefore, we suspect that, in Ref. 20, the P₄ precursors have to be either first decomposed into P₂ dimers, then fast feed to the growing islands, or have to be supplied at a much slower rate as P₄ entities (see, for example, New. J. Phys. 10, 093026 (2008) & 11, 063046 (2009) for chain feeding in the case of graphene). Based on these calculations, we may conclude that two P atoms prefer to form a stable dimer on Cu(111) or P₂ dimers can be directly deposited on Cu(111), and such dimers are energetically stable and can diffuse much faster than either a P monomer or (especially) a P₄ cluster. Thus, the growth precursor (P₂ in the present case) in epitaxial growth may play a decisive role in the formation of ultraflat BlueP on Cu(111),

as presented in the present study.”

Figure. R6 Top (upper panel) and side (lower panel) views of the most stable configurations of (a) a P monomer, (b) a P-P dimer, and (c) two separated P adatoms on the Cu(111) surface.

Figure. R7 a, Diffusion pathways of a P monomer on Cu(111), with corresponding energy profiles presented in (b). c-d and e-f depict the same information as (a) and (b), respectively, but for the cases of an adsorbed P-P dimer and P₄. The solid and dotted blue balls represent the initial and final positions of the P adatoms, respectively.

Comments 2: According to my Comment 3 and 6 (also in relation to Fig. S2), mentioned applications about chirality sensitive spintronics and asymmetric catalysis are indeed interesting as perspectives but they should be grounded on a compliant stabilisation scheme without which the highly unstable blue-phosphorene turns out to be unpractical. In addition to the work by Li et al. Nature Nanotechnol. 10, 227 (2015) as reported in the Response, let me suggest the Authors to optionally consider material selective scheme for encapsulation like those implemented in case of silicene by Molle et al, Faraday Discuss. 227, 171 (2021), black

Phosphorus by Wood et al, Nano Lett. Nano Lett. 14, 6964 (2014), or blue-phosphorene on Au by Grazianetti et al, Nanoscale 11, 18232 (2019) as they may offer viable solutions of material exploitation out of the vacuum condition of the growth environment.

Responses: We thank the reviewer for the helpful suggestions. As revealed by XPS (see Fig. S2), the chiral BlueP is indeed vulnerable to environmental degradation; therefore, proper encapsulation must be considered in potential future practical applications. As reported in the literature, in the case of silicene [Nature Nanotechnol. 10, 227 (2015) and Faraday Discuss. 227, 171 (2021)], black Phosphorus [Nano Lett. 14, 6964 (2014)], and blue phosphorene on Au(111) [Nanoscale 11, 18232 (2019)], several experimental groups have realized effective encapsulation by capping Al₂O₃ or AlO_x on the films to prevent the sample from directly exposing to the air. As suggested by the reviewer, these effective encapsulation schemes could also be exploited in future studies of chiral BlueP.

In response to the Reviewer's comment, on page 9 in the main text, the description about encapsulation of the chiral BlueP **“therefore, for some practical application, proper encapsulation is needed.”** has been expanded to **“therefore, proper encapsulation schemes such as those implemented in the cases of silicene [Nature Nanotechnol. 10, 227 (2015) and Faraday Discuss. 227, 171 (2021), cited as Ref-47 and Ref-48], black Phosphorus [Nano Lett. 14, 6964 (2014), cited as Ref-49], and blue-phosphorene on Au [Nanoscale 11, 18232 (2019), cited as Ref-50], could be exploited in future explorations of its potential practical applications.”**

Generic Comments: *On top of these remarks, I think the manuscript extensively reports on a new fashion of temperature controllable phosphorene phases as never done before. It is not clear how to make use of these phosphorene allotropes in practical applications, nonetheless here one can find the details on how to produce a chiral phosphorene in a feasible processing scheme for further progress on the emerging chirality-sensitive materials applications. Therefore, I am pleased to consider the paper as suitable for dissemination through Nature Communications after the Authors will be able to address my present Comment 1 more specifically.*

Responses: We are surely also pleased that the reviewer finds this work suitable for dissemination through Nature Communications.

Finally, we wish to sincerely thank all the three reviewers again for their careful and expertized reviews, as well as for their constructive comments and suggestions. We hope this further revised manuscript is now ready for publication in Nature Communications.

REVIEWERS' COMMENTS

Reviewer #1 (Remarks to the Author):

The authors have answered all my comments properly. They have also revised the manuscript and added new data. The manuscript is significantly improved. I am happy to recommend its acceptance.

Reviewer #3 (Remarks to the Author):

The Authors have extensively replied to my comments with additional arguments and convincing discussions (also in consideration of other reviewers' comments) thus improving and consolidating their work that I consider suitable for publication in Nature Communications as is.